# Cap-independent co-expression of dsRNA-sensing and NF-κB pathway inhibitors enables controllable self-amplifying RNA expression with reduced immunotoxicity

**Tony KY Lim[1]\*, Anne Ritoux[1], Luke W Paine[1], Larissa Ferguson[2], Tawab Abdul[1], Laura J Grundy[1], Ewan St John Smith[1]\***

[1]Department of Pharmacology, University of Cambridge, Cambridge, United Kingdom; [2]MRC Laboratory of Molecular Biology, Cambridge, United Kingdom

## eLife Assessment

In this manuscript, Lim and collaborators present an **important** system for developing self-amplifying RNA with **convincing** evidence that it does not provoke a strong host inflammatory response in cultured cells. This approach could be further strengthened going forward by testing these self-amplying RNAs in an in vivo system.

**\*For correspondence:**
tony.ky.lim@gmail.com (TKYL);
es336@cam.ac.uk (ESJS)

**Competing interest:** The authors declare that no competing interests exist.

**Abstract** Self-amplifying RNA (saRNA) holds promise for durable therapeutic gene expression, but its broader utility beyond vaccines is limited by potent innate immune responses triggered during replication. These responses shut down translation, induce cytotoxicity, degrade host mRNAs, and drive cytokine production. While exogenous immunosuppressants can blunt these effects, they complicate treatment and risk systemic side effects. To address this, we engineered 'immune-evasive saRNA' that intrinsically suppresses the innate immune pathways triggered by its own replication. This strategy leverages cap-independent translation to co-express a suite of inhibitors from a single saRNA transcript, targeting key innate immune pathways, including protein kinase R (PKR), oligoadenylate synthase (OAS)/RNase L, and nuclear factor-κB (NF-κB). In primary mouse fibroblast-like synoviocytes, a cell type central to the pathology of joint diseases, immune-evasive saRNA enables sustained transgene expression without external immunosuppressants, substantially reducing cytotoxicity and antiviral cytokine secretion. Crucially, this system offers both concentration-dependent control of expression and on-demand termination via a small-molecule antiviral. Together, these findings establish a framework for developing saRNA therapeutics with an improved tolerability profile that can be switched off once therapeutic outcomes are met, offering a path toward a controllable gene expression platform that fills the therapeutic gap between the transience of mRNA and the permanence of viral vectors.

## Introduction

Self-amplifying RNA (saRNA) is a promising platform for therapeutic gene expression due to its ability to encode large genetic payloads and achieve sustained, non-integrating protein expression (*Frolov et al., 1996*). Upon cellular internalization, saRNA utilizes host machinery to synthesize an RNA-dependent RNA polymerase (RdRp), which drives replication of the saRNA genome (*Bloom*

*et al., 2021*). This process involves the production of a negative-strand RNA intermediate that serves as a template for both genomic and subgenomic RNA synthesis (*Pietilä et al., 2017*). Subgenomic RNA, encoding the gene of interest, is transcribed in excess of the genomic RNA, leading to robust and sustained protein expression (*Maruggi et al., 2019*). However, the self-replication of saRNA also generates double-stranded RNA (dsRNA) intermediates, formed both during the synthesis of negative-strand RNA and during the transcription of positive-strand RNA from the negative-strand template (*Pietilä et al., 2017*). These dsRNA intermediates are potent activators of cytosolic pattern recognition receptors, triggering innate immune responses (*Akhrymuk et al., 2016*; *Gong et al., 2024*). This recognition results in the shutdown of cap-dependent translation (*Gorchakov et al., 2004*; *Dominguez et al., 2023*; *Frolov and Schlesinger, 1994*), degradation of cellular mRNA (*Gong et al., 2024*; *Terenzi et al., 1999*), induction of programmed cell death (*Venticinque and Meruelo, 2010*; *Frolov et al., 1999*; *Lundstrom et al., 2003*), and release of cytokines and chemokines characteristic of viral infection (*Vanluchene et al., 2024*; *Pepini et al., 2017*; *Blakney et al., 2021b*). In non-immunotherapeutic applications such as gene or protein therapy, these responses are particularly problematic as they limit transgene expression, reshape the host transcriptional landscape in ways that may undermine therapeutic effects, deplete transfected cells, and induce inflammation.

Strategies to mitigate saRNA-mediated innate immune responses have included the incorporation of modified nucleotides (*McGee et al., 2025*), removal of dsRNA contaminants (*Vanluchene et al., 2024*; *Zhong et al., 2021*), and co-delivery of non-replicating mRNA encoding viral innate immune inhibiting proteins (*Beissert et al., 2017*; *Yoshioka et al., 2013*; *Xue et al., 2024*). While these approaches can reduce initial innate immune responses to saRNA, they fall short in addressing the immune activation triggered by dsRNA intermediates that arise during saRNA replication (*Bloom et al., 2021*). Sequence evolution of the RdRp shows some promise in reducing these responses, but even engineered RdRps still induce significant innate immune activation (*Gong et al., 2024*; *Li et al., 2019*). Encoding viral immune inhibitors within the saRNA construct using 2A self-cleaving peptides can improve protein expression in vitro but has limited impact on cytokine responses, despite reductions in NF-κB and IRF3 activation (*Blakney et al., 2021b*). Although these strategies temper saRNA-induced innate immune responses to levels compatible with immunotherapy—where some immune activation is beneficial (*Blakney et al., 2021a*)—they fall short of the stringent requirements for non-immunotherapeutic contexts, where immune activation can compromise therapeutic outcomes. In such cases, exogenous immunosuppressants can attenuate saRNA-induced immune responses to enable effective transgene expression (*Vanluchene et al., 2024*; *Yoshioka et al., 2013*; *Erasmus et al., 2020*; *Kim et al., 2017*), but this approach complicates treatment regimens and increases the risk of unintended side effects from systemic immune suppression.

To address these challenges, we developed a fully saRNA-based approach that mitigates innate immune responses triggered by saRNA replication, incorporating several key innovations. First, innate immune inhibitory proteins are expressed via cap-independent translation, bypassing the cap-dependent translation shutdown commonly triggered by saRNA. Cap-dependent translation requires the presence of a 5′ cap structure on mRNA, which recruits translation initiation factors and ribosomes (*Merrick, 2004*). By contrast, cap-independent translation uses internal ribosome entry sites (IRES) to recruit ribosomes directly to the mRNA, enabling translation even when cap-dependent pathways are inhibited (*Yang and Wang, 2019*). Second, by simultaneously targeting multiple dsRNA-sensing and inflammatory signaling pathways, our method provides a more comprehensive suppression of innate immune responses than single-target approaches. Third, encoding innate immune inhibitors directly within the saRNA ensures continuous protection against persistent innate immune activation caused by dsRNA intermediates during saRNA replication. We hypothesized that this cap-independent, multi-pronged, cis-encoded strategy would yield an 'immune-evasive' saRNA platform capable of sustained transgene expression with reduced immunotoxicity.

We evaluated this approach in primary mouse fibroblast-like synoviocytes (FLS), a key cell type in the pathology of joint diseases (*Maglaviceanu et al., 2021*; *Mousavi et al., 2021*). Using a novel microplate assay for longitudinal monitoring of saRNA translation control and cytotoxicity in live cells, we demonstrate that this strategy effectively reduces saRNA-induced cytotoxicity and cytokine production while enabling long-term transgene expression that does not require external immunosuppressants. This approach also enables external control of transgene expression with a small-molecule antiviral, allowing reversible or irreversible suppression depending on concentration, thereby offering

both dosing flexibility and a mechanism for replicon removal once therapeutic goals are met. These findings establish a foundation for developing saRNA-based therapeutics that are durable, well-tolerated, and externally controllable for applications beyond vaccines.

## Results

### A microplate assay for longitudinal monitoring of subgenomic- and IRES-driven transgene expression and cell number

Because saRNA can shut down cap-dependent translation and induce cytotoxicity, we developed a microplate-based assay to simultaneously monitor subgenomic (cap-dependent) and IRES-mediated (cap-independent) transgene expression, together with cell number. Detailed assay development—including correction for spectral overlap and validation of BioTracker NIR680 as a cell number proxy—is provided in Appendix 1. To distinguish between translation initiation mechanisms, we designed dual-fluorescence saRNA reporter constructs (*Figure 1a*): mScarlet3 was expressed from the subgenomic RNA via cap-dependent initiation, while EGFP was expressed from an encephalomyocarditis virus (EMCV) IRES via cap-independent initiation. This assay permits repeated, non-destructive measurement of saRNA-driven transgene expression and cell number in primary mouse FLS, enabling longitudinal analysis of the effects of embedding inhibitors of innate immune pathways within saRNA.

### Design of saRNA constructs for inhibiting multiple dsRNA-sensing pathways

To attenuate saRNA-induced innate immune responses, we designed constructs that inhibit dsRNA-sensing pathways. Cells detect cytosolic dsRNA through several pathways, including RIG-I (*Rehwinkel and Gack, 2020*), melanoma differentiation-associated protein 5 (MDA-5) (*Dias Junior et al., 2019*), protein kinase R (PKR) (*Gal-Ben-Ari et al., 2018*), and oligoadenylate synthase (OAS)/RNase L pathways (*Silverman, 2007*). To broadly inhibit these pathways, we designed an saRNA construct, referred to as 'E3', that co-expresses the vaccinia virus E3 protein (*Figure 1a*). E3 protein is a pleiotropic inhibitor that binds and sequesters dsRNA, effectively blocking multiple dsRNA-sensing pathways (*Szczerba et al., 2022*).

Given the key roles of PKR in cap-dependent translation shutdown and OAS/RNase L in cellular mRNA degradation in response to dsRNA, we sought to further inhibit these dsRNA-sensing pathways. We designed a second construct that, in additon to co-expressing vaccinia virus E3, includes Toscana virus NSs and Theiler's virus L*. Toscana virus NSs is a ubiquitin ligase that promotes PKR degradation (*Kalveram and Ikegami, 2013*), while Theiler's virus L* is an inhibitor of mouse—but not human—RNase L (*Drappier et al., 2018*; *Sorgeloos et al., 2013*). To enable co-expression of E3, NSs, and L*, we incorporated 2A 'self-cleaving' peptide sequences between the proteins, allowing the polyprotein to be cleaved into separate proteins during translation. As prior experiments showed that using multiple identical 2A peptides can reduce protein expression (*Liu et al., 2017a*), we utilized nonidentical 2A peptides. This construct, named 'E3-NSs-L*', provides a more comprehensive inhibition of dsRNA-sensing pathways than the E3 construct.

Because translation shutdown is one of the major cellular responses to saRNA replication, we expressed these dsRNA-sensing pathway inhibitors using ECMV IRES-mediated, cap-independent translation. During translation shutdown, global cap-dependent mRNA translation initiation is limited, freeing ribosomes that would otherwise be used for translating capped mRNAs (*Baird and Wek, 2012*), thereby enhancing cap-independent translation (*Koch et al., 2020*). By leveraging this mechanism, IRES-mediated expression of dsRNA-sensing pathway inhibitors ensures their levels increase as translation shutdown intensifies. In contrast, encoding these inhibitors via cap-dependent translation is expected to be less effective, as translation shutdown would lead to reduced inhibitor expression.

As a control, we designed a conventional saRNA construct that expresses a blue fluorescent protein (moxBFP) as a visual marker of IRES functionality in place of inhibitors of dsRNA-sensing pathways. Importantly, moxBFP expression did not overlap spectrally with the EGFP or mScarlet3 channels (*Appendix 1—figure 3b*).

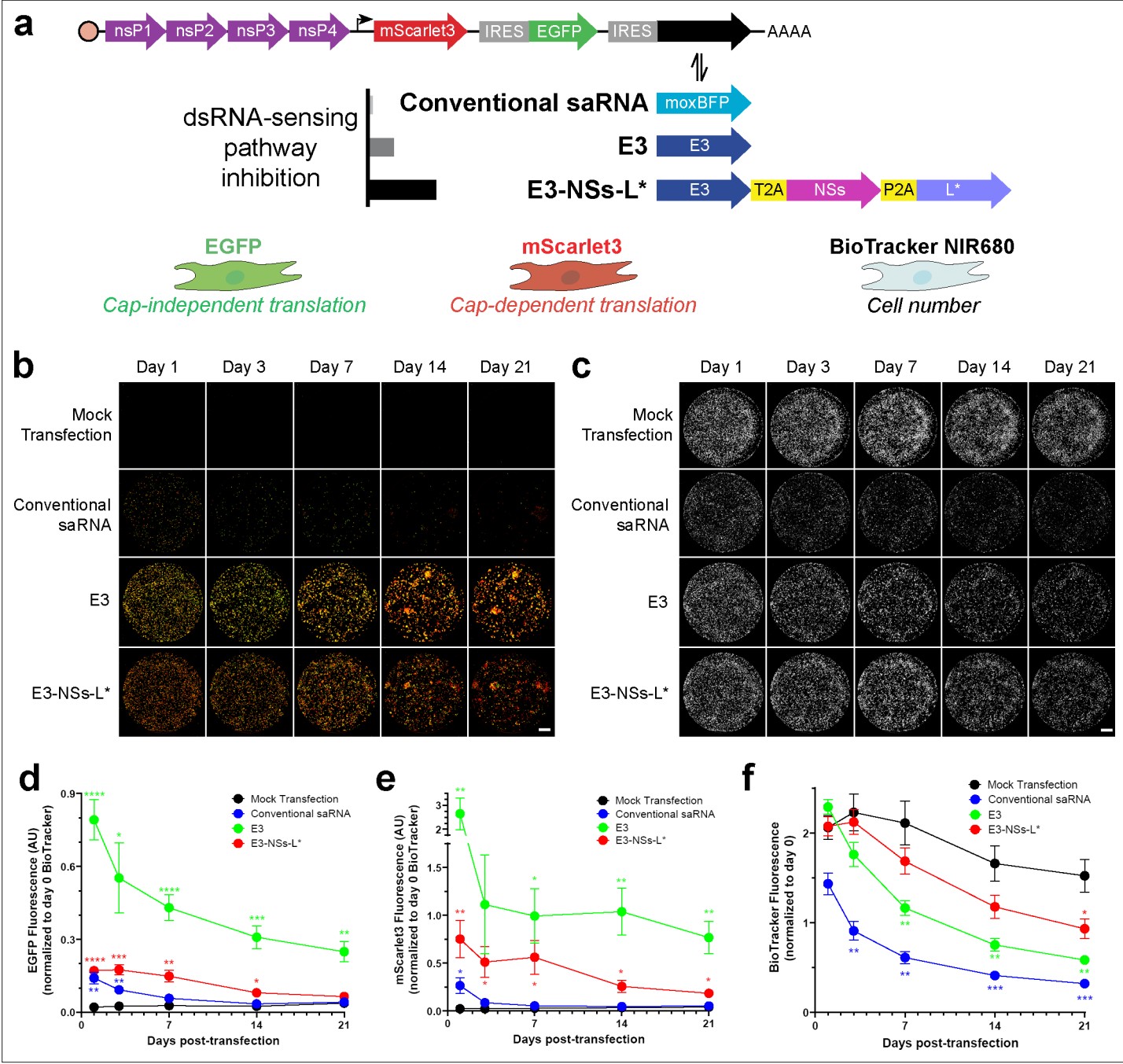

**Figure 1.** Differential effects of moderate and strong double-stranded RNA (dsRNA)-sensing pathway inhibition on self-amplifying RNA (saRNA) transgene expression and cell number. (**a**) Schematic of saRNA constructs co-expressing fluorescent reporters and dsRNA-sensing pathway inhibitors. nSP1–4 encodes the saRNA replicase. mScarlet3 indicates cap-dependent transgene expression and is expressed from subgenomic RNA (angled arrow denotes subgenomic promoter). EGFP indicates cap-independent transgene expression and is expressed from an IRES. A second IRES expresses varying levels of dsRNA-sensing pathway inhibition: 'Conventional saRNA' expresses moxBFP (control). 'E3' expresses vaccinia virus E3 (dsRNA-binding protein). 'E3-NSs-L*' expresses E3 plus Toscana virus NSs (ubiquitin ligase targeting PKR [protein kinase R]) and Theiler's virus L* (RNase L inhibitor). saRNA was transfected into mouse primary fibroblast-like synoviocytes (FLS), labeled with BioTracker to indicate cell number. (**b**) Representative composite images of microplate wells showing EGFP (green) and mScarlet3 (red). (**c**) Representative images of microplate wells showing BioTracker. (**d**) Longitudinal quantification of EGFP (n=11). E3 provided the highest expression, while E3-NSs-L* expressed at intermediate levels. (**e**) Longitudinal quantification of mScarlet3 (n=11). E3 provided the highest expression, while E3-NSs-L* expressed at intermediate levels. (**f**) Longitudinal quantification of BioTracker (n=11). Conventional saRNA led to immediate and long-term reductions in signal. E3 initially maintained signal, but it gradually decreased over time. E3-NSs-L* preserved signal throughout the time course. For panels (b–c): Scale bar = 5 mm. For panels (d–f): Data are normalized to starting cell number (pre-transfection BioTracker signal). Statistical significance relative to mock transfection was assessed using two-way repeated-measures

*Figure 1 continued on next page*

*Figure 1 continued*

(RM) ANOVA with Greenhouse-Geisser correction and Dunnett's multiple comparisons test. *p<0.05, **p<0.01, ***p<0.001, and ****p<0.0001. Data are presented as mean ± standard error of the mean (SEM). The mock transfection control data is also presented in *Figure 5c–e*.

The online version of this article includes the following source data and figure supplement(s) for figure 1:

Source data 1. Whole-plasmid sequencing results for self-amplifying RNA (saRNA) constructs depicted in *Figure 1a*.

Source data 2. Numerical data used to generate the plots in *Figure 1*.

Figure supplement 1. Inhibiting dsRNA-sensing pathways or saRNA replication mitigates saRNA-induced cell loss.

Figure supplement 1—source data 1. Numerical data used to generate the plots in *Figure 1—figure supplement 1*.

## Moderate dsRNA-sensing pathway inhibition enables high transgene expression at the cost of cell loss, while strong inhibition preserves cell number at lower expression levels

Following transfection of saRNA constructs, we monitored BioTracker, EGFP, and mScarlet3 fluorescence over 3 weeks (*Figure 1b and c*). Inhibiting dsRNA-sensing pathways with viral proteins significantly enhanced saRNA transgene expression (*Figure 1d and e*). Interestingly, the E3 construct produced the highest levels of EGFP and mScarlet3 expression, surpassing both conventional saRNA and the E3-NSs-L* construct. Conventional saRNA yielded low levels of transgene expression, while E3-NSs-L* showed intermediate expression.

Achieving durable transgene expression requires not only high expression but also preservation of cell viability, as constructs that induce cell death undermine this goal. Transfection of FLS with conventional saRNA caused both immediate and long-term reductions in cell number, as indicated by BioTracker (*Figure 1f*). While the E3 construct initially maintained cell number, it gradually diminished over time. In contrast, the E3-NSs-L* construct provided sustained protection against this decline. A time-integrated analysis of these data (*Figure 1—figure supplement 1a*) and CellTag-based normalization assays on day 2 post-transfection (*Figure 1—figure supplement 1b*) further supported these results, revealing a stepwise increase in protection against cell loss with greater dsRNA-sensing pathway inhibition across the three constructs. Together, these results indicate that while the E3 construct enhances saRNA transgene expression, it also induces cell loss, underscoring the advantage of E3-NSs-L* for applications requiring sustained, non-cytotoxic gene expression.

## Inhibiting saRNA replication mitigates saRNA-induced cytotoxicity

In vitro transcription of saRNA often generates dsRNA byproducts, which can activate innate immune pathways and contribute to cytotoxicity (*Zhong et al., 2021*; *Baiersdörfer et al., 2019*). It remains unclear whether saRNA-induced cell loss arises primarily from these byproducts or from cytosolic dsRNA generated during saRNA replication. To investigate this, FLS were mock transfected or transfected with conventional saRNA in the presence or absence of 10 µM ML336, a potent inhibitor of the Venezuelan equine encephalitis virus (VEEV) RdRp that mediates self-amplification (*Chung et al., 2010*). Without ML336, conventional saRNA-transfected cultures exhibited reduced viability as measured by calcein AM—a viability dye that live cells convert into a fluorescent signal (*Bratosin et al., 2005*). In contrast, blocking saRNA replication with ML336 restored cell viability to levels comparable to mock transfection (*Figure 1—figure supplement 1c*). These findings suggest that saRNA-induced cell loss is primarily driven by dsRNA generated during replication, rather than by dsRNA byproducts formed during in vitro transcription.

## Inhibiting dsRNA-sensing pathways reduces saRNA-induced cytotoxicity and improves cell viability

saRNA is known to induce programmed cell death (*Frolov and Schlesinger, 1994*; *Li et al., 2020*; *Mastrangelo et al., 2000*), which often leads to cell detachment (*Costigan et al., 2023*). Reductions in BioTracker signal are likely indicative of this process. However, since BioTracker is a lipophilic membrane dye, the reduction in signal may not necessarily correlate with cell viability or cytotoxicity. In addition to long-term cell tracking, lipophilic membrane dyes like BioTracker are often used to stain extracellular vesicles, which bud off from the plasma membrane (*Liu et al., 2022*). During cellular stress, increased production of extracellular vesicles (*Chiaradia et al., 2021*) could deplete

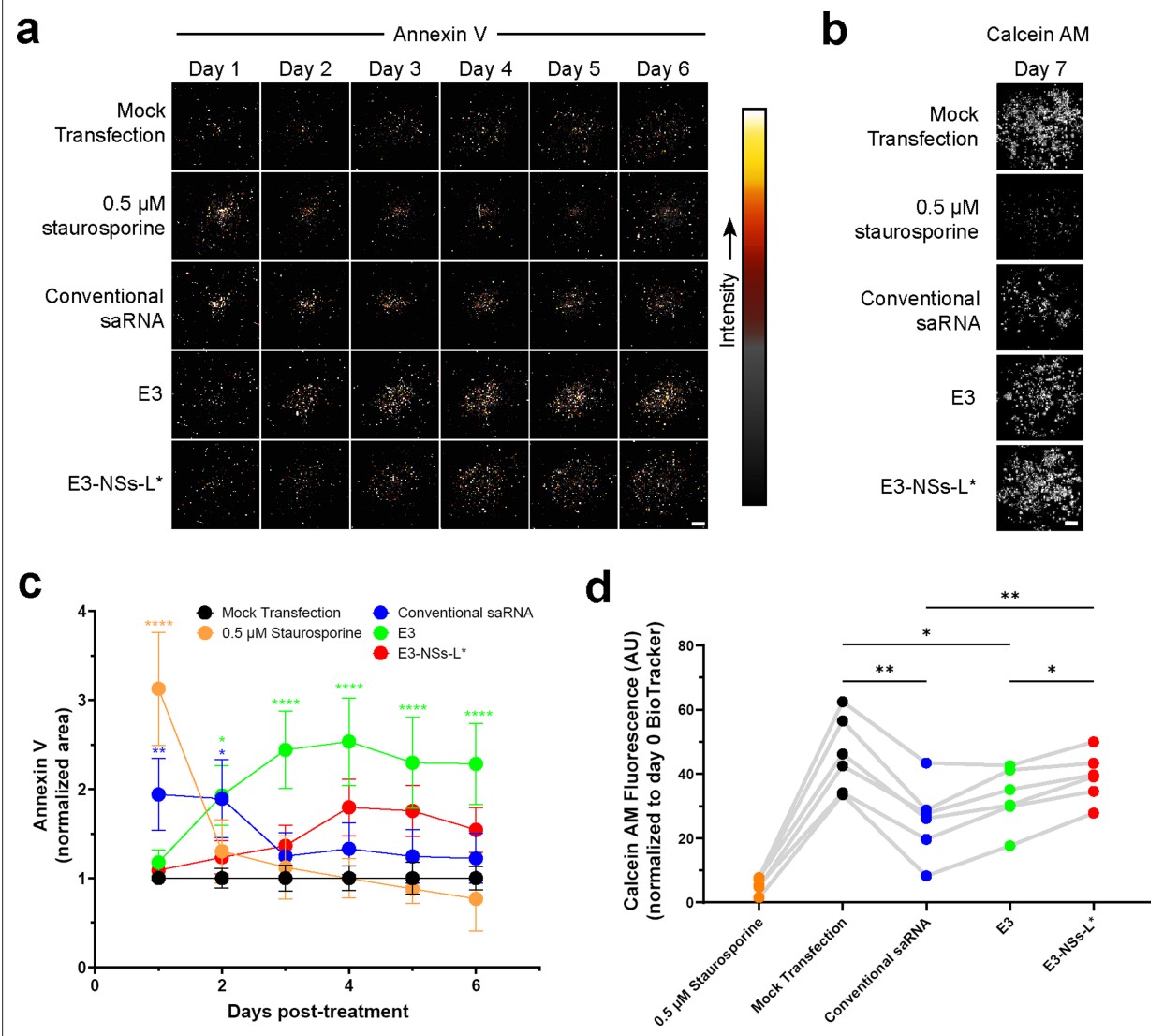

**Figure 2.** E3-NSs-L* prevents saRNA-induced elevations in annexin V staining and reductions in cell viability. (**a**) Representative cropped microplate well images of Annexin V-CF800, indicating phosphatidylserine exposure or loss of membrane integrity. FLS were transfected with saRNA or treated with staurosporine, an apoptosis inducer. (**b**) Representative cropped microplate well images of calcein AM. Cultures are the same as in panel (a). (**c**) Quantification of Annexin V positive area, determined using Li thresholding and normalized to average mock transfection values (n=6). Annexin V positive area was increased by staurosporine, conventional saRNA, and E3—but not by E3-NSs-L*. Statistical significance relative to mock transfection was determined using two-way RM ANOVA with Bonferroni's multiple comparisons test. Data are presented as mean ± SEM. (**d**) Quantification of calcein AM (n=6). Conventional saRNA and E3 reduced signal, an effect not observed with E3-NSs-L*. Data are normalized to starting cell number (pre-transfection BioTracker signal). Statistical significance was determined by one-way RM ANOVA with Greenhouse-Geisser correction and Tukey's multiple comparisons test comparing all groups. All groups differed significantly from staurosporine (significance indicators omitted for clarity). Connecting lines indicate responses from the same biological replicate. For panels (a–b): Scale bar = 1.5 mm. For panels (c–d): *p<0.05, **p<0.01, ***p<0.001, and ****p<0.0001.

The online version of this article includes the following source data for figure 2:

**Source data 1.** Numerical data used to generate the plots in *Figure 2*.

the membrane-associated BioTracker dye from cells, resulting in a lower detectable signal. Therefore, the observed reduction in BioTracker signal could reflect enhanced extracellular vesicle production rather than cell death.

To provide more definitive evidence that inhibiting dsRNA-sensing pathways protects against saRNA-induced cytotoxicity, we conducted a longitudinal assay with annexin V staining over 6 days (*Figure 2a*), followed by calcein AM staining on day 7 (*Figure 2b*). Annexin V, a membrane-impermeable

protein, binds to phosphatidylserine, which translocates to the extracellular side of the cell membrane during early apoptosis, enabling the detection of apoptotic cells with intact membranes (*van Genderen et al., 2006*). If the membrane is compromised, annexin V can enter cells and stain intracellular phosphatidylserine, indicating late apoptosis or necrosis (*Crowley, 2016*). Calcein AM, on the other hand, accumulates in metabolically active cells with intact membranes, serving both as a marker of live cell number and an indicator of membrane integrity (*Chan et al., 2012*).

The apoptosis-inducing agent staurosporine caused a transient increase in annexin V staining that decreased after the first day (*Figure 2c*), along with a marked reduction in calcein AM fluorescence compared to mock transfection (*Figure 2d*), consistent with extracellular phosphatidylserine translocation and/or increased membrane permeability, followed by cell detachment. Similarly, conventional saRNA induced a temporary increase in annexin V staining during the first 2 days post-transfection (*Figure 2c*) accompanied by reduced calcein AM staining (*Figure 2d*). Co-expression of E3 prevented the initial increase in annexin V staining but elevated it from day 2 onward and also resulted in significantly reduced calcein AM intensity (*Figure 2c and d*). In contrast, E3-NSs-L* had no significant effect on annexin V or calcein AM staining compared to mock transfection, supporting the idea that broader inhibition of dsRNA-sensing pathways, achieved by combining E3, NSs, and L*, is more effective than E3 alone in preventing saRNA-induced cell death and preserving cell viability.

## dsRNA-sensing pathway inhibition prevents eIF2α phosphorylation, but fails to affect saRNA-induced reductions in eIF4E phosphorylation levels

After establishing the effects of moderate and strong inhibition of dsRNA-sensing pathways on cytotoxicity and transgene expression, we next investigated how modulating these pathways influences translational control at the molecular level. Eukaryotic translation initiation factor (eIF)2α is a central regulator of translation initiation; its phosphorylation serves as a cellular stress response to limit cellular cap-dependent protein synthesis (*Baird and Wek, 2012*). Using in-cell western assays, we found that conventional saRNA increased eIF2α phosphorylation, but both E3 and E3-NSs-L* effectively blocked this phosphorylation (*Figure 3a*), consistent with previous reports of vaccinia virus E3's inhibitory effect on eIF2α phosphorylation (*Beissert et al., 2017*; *Xue et al., 2024*). Interestingly, E3-NSs-L* also increased total eIF2α levels, an effect not observed with the E3 construct (*Figure 3b*).

We next examined the impact of saRNA on eIF4E, another key regulator of translation initiation (*Mars et al., 2024*). Unlike eIF2α phosphorylation—which was inhibited by the E3 and E3-NSs-L* constructs—phosphorylation of eIF4E was reduced by all saRNA constructs (*Figure 3c*), including those targeting dsRNA-sensing pathways, while total eIF4E levels remained unchanged (*Figure 3d*). These results indicate that inhibiting dsRNA-sensing pathways does not prevent the saRNA-induced reduction in eIF4E phosphorylation.

Given that PKR phosphorylates eIF2α upon activation by dsRNA (*Gorchakov et al., 2004*), we next assessed PKR levels following saRNA transfection. Consistent with previous reports (*Gong et al., 2024*), transfection with conventional saRNA led to increased PKR levels (*Figure 3—figure supplement 1a*). The E3 construct produced a similar increase, whereas co-expression of E3-NSs-L* mitigated this effect. This finding aligns with the established role of the Toscana virus NSs protein as a ubiquitin ligase that targets PKR for degradation (*Kalveram and Ikegami, 2013*).

## Host mRNA degradation contributes to enhanced transgene expression under moderate dsRNA-sensing pathway inhibition

The increased transgene expression observed with the E3 construct compared to E3-NSs-L* cannot be attributed to differences in eIF2α or eIF4E-mediated translational control. Phosphorylation levels of both factors were comparable between the two constructs (*Figure 3a and c*), indicating that the enhanced transgene expression associated with the E3 construct arises from alternative mechanisms.

Upon detection of dsRNA, OAS enzymes synthesize 2′–5′-oligoadenylates, which activate RNase L, a ribonuclease that degrades single-stranded RNA. This leads to widespread host mRNA cleavage (*Burke et al., 2019*), impaired nuclear export of transcripts (*Burke et al., 2021*), and global arrest of protein synthesis (*Donovan et al., 2017*). Several RNA viruses, including dengue and Zika, exploit this process by localizing their transcripts within replication organelles, shielding them from RNase L activity while maintaining efficient translation (*Whelan et al., 2019*; *Burke et al., 2021*). The VEEV-derived

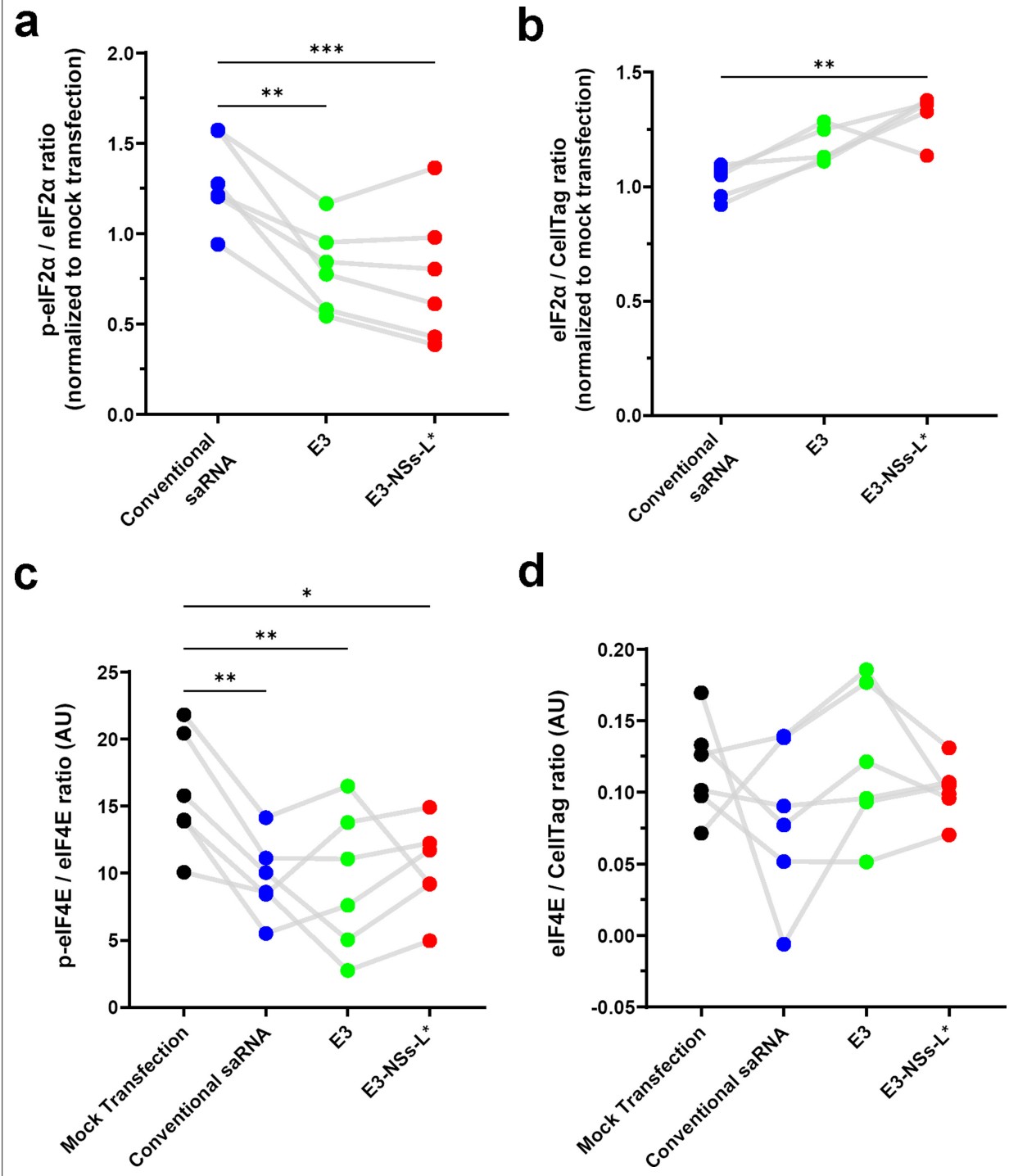

**Figure 3.** dsRNA-sensing pathway inhibitors suppress saRNA-induced eIF2α phosphorylation but not the reduction in eIF4E phosphorylation. (**a**) Phosphorylated eIF2α levels examined by in-cell western (n=6). Conventional saRNA increased eIF2α phosphorylation, while E3 and E3-NSs-L* did not. Data are normalized to total eIF2α and presented as fold-change relative to mock transfection. (**b**) Total eIF2α levels examined by in-cell western (n=5). E3-NSs-L* increased total eIF2α. Data are normalized to CellTag and presented as fold-change relative to mock transfection. (**c**) Phosphorylated eIF4E levels, normalized to total eIF4E, examined by in-cell western (n=6). All constructs reduced eIF4E phosphorylation. (**d**) Total eIF4E levels, normalized to CellTag, examined by in-cell western (n=6). No significant differences were revealed ($F_{(3,15)} = 1.207$, $p=0.3410$). For all panels: Statistical significance was determined by one-way RM ANOVA and Tukey's multiple comparison test. *$p<0.05$, **$p<0.01$, ***$p<0.001$. The mock transfection control data are also used in **Figure 6a–d**. Connecting lines indicate responses from the same biological replicate.

The online version of this article includes the following source data and figure supplement(s) for figure 3:

*Figure 3 continued on next page*

*Figure 3 continued*

**Source data 1.** Numerical data used to generate the plots in *Figure 3*.

**Figure supplement 1.** E3-NSs-L* protects against saRNA-induced PKR upregulation, RNA degradation, and decreased protein synthesis rate.

**Figure supplement 1—source data 1.** Numerical data used to generate the plots in *Figure 3—figure supplement 1*.

saRNA used in this study also replicates within these organelles (*Paul and Bartenschlager, 2013*), likely protecting its transcripts from RNase L-mediated degradation in a similar fashion.

We hypothesized that RNase L activation by the E3 construct would preferentially deplete host mRNAs, reducing competition for the ribosomal machinery and thereby enhancing the relative translation of protected saRNA transcripts. In contrast, E3-NSs-L* expresses Theiler's virus L*, an RNase L inhibitor (*Drappier et al., 2018*), preventing host mRNA degradation and forcing saRNA to compete with host mRNA for ribosome access.

To evaluate whether RNase L activation contributes to the greater transgene expression observed with the E3 construct compared to E3-NSs-L*, we assessed ribosomal RNA (rRNA) integrity as a surrogate measure of RNase L activity (*Silverman, 2007*). For this analysis, total RNA was extracted from FLS transfected with either E3 or E3-NSs-L*; the conventional saRNA construct was not examined, as its induction of translation shutdown is expected to alter dsRNA intermediate levels. rRNA integrity was then evaluated using the RNA integrity number (RIN) algorithm on capillary gel electropherograms of total RNA (*Schroeder et al., 2006*). Cells transfected with E3 exhibited a moderate but statistically significant reduction in RIN compared to those transfected with E3-NSs-L* (*Figure 3—figure supplement 1b*). This finding aligns with previous studies demonstrating that RNase L activation primarily depletes host mRNA levels before extensive rRNA degradation (*Donovan et al., 2017*; *Burke et al., 2019*; *Karasik et al., 2021*), and supports the hypothesis that RNase L activation and subsequent reductions in host mRNA occur with the E3 construct but not with E3-NSs-L*.

We further evaluated global protein synthesis using surface sensing of translation (SUnSET) (*Ravi et al., 2020*; *Schmidt et al., 2009*), in which FLS were pulsed with puromycin, and newly synthesized peptides were detected via anti-puromycin immunoreactivity (*Figure 3—figure supplement 1c*). Transfection with conventional saRNA led to a marked reduction in protein synthesis, consistent with translation shutdown. Co-expression of E3 partially restored translation; however, synthesis rates remained significantly lower than in mock-transfected cells, likely due to RNase L-mediated depletion of host mRNAs despite relief from translation shutdown. In contrast, transfection with E3-NSs-L* resulted in protein synthesis rates comparable to those of mock-transfected cells, consistent with effective inhibition of RNase L and preservation of host mRNA.

Together, these findings support a model in which RNase L activation by the E3 construct enhances transgene expression through selective depletion of host mRNA, thereby reducing competition for the ribosomal pool. Conversely, E3-NSs-L* suppresses RNase L activity, preserving host mRNA and limiting the availability of ribosomes for transgene expression.

## dsRNA-sensing pathway inhibition suppresses secretion of select antiviral cytokines while enhancing others

For therapeutic applications beyond vaccines and cancer immunotherapy, prolonged transgene expression is ideally achieved without triggering immune activation. While our data show that inhibition of dsRNA-sensing pathways can mitigate key saRNA-induced adverse effects, including cell death, translation shutdown, and host mRNA degradation, it remains unclear whether this strategy also attenuates cytokine secretion. To investigate this, we measured the concentrations of 13 antiviral cytokines in FLS culture supernatants using a multiplex bead-based immunoassay (*Figure 4a*, *Figure 4—figure supplement 1*). Transfection with conventional saRNA elicited a broad cytokine response. Inhibition of dsRNA-sensing pathways selectively suppressed production of certain cytokines: Both E3 and E3-NSs-L* significantly reduced interferon (IFN)-α and IFN-β, while E3-NSs-L* significantly reduced tumor necrosis factor (TNF). Notably, the production of some cytokines was elevated: E3 significantly increased macrophage chemoattractant protein-1, and E3-NSs-L* significantly increased granulocyte-macrophage colony-stimulating factor, relative to mock-transfected controls. These findings indicate that even strong inhibition of dsRNA-sensing pathways only partially

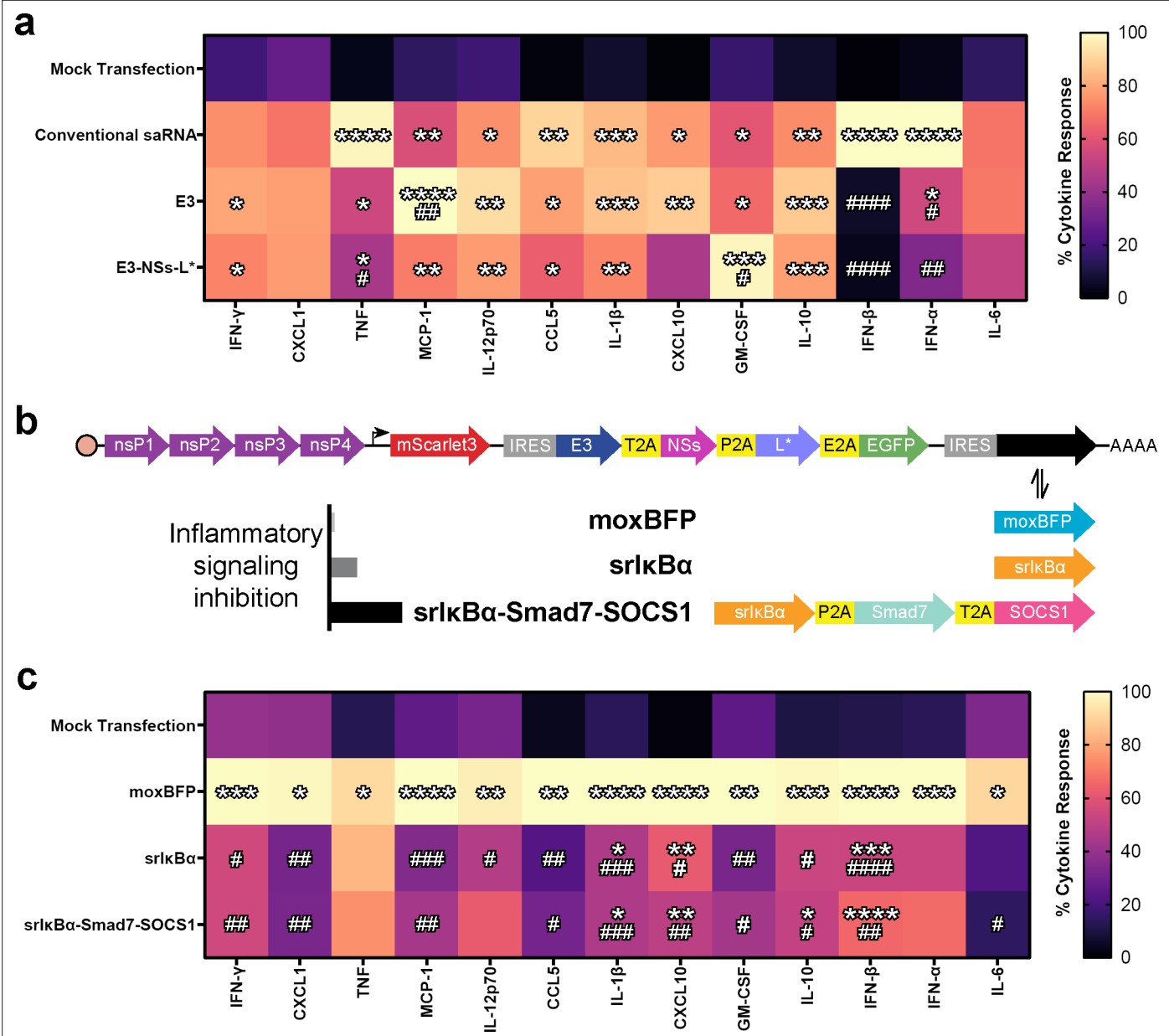

**Figure 4.** saRNA-triggered cytokine responses are variably affected by dsRNA sensing inhibition, but broadly suppressed by NF-κB inhibition. (**a**) Antiviral cytokine response quantified by multiplex bead-based immunoassay (n=6). Inhibition of dsRNA-sensing pathways reduced some cytokine responses while enhancing others. Cytokine levels were normalized to pre-transfection cell number (indicated by BioTracker), scaled within each biological replicate with the highest value set to 100%, and shown as a heatmap of group means. Plots of unscaled data are presented in *Figure 4—figure supplement 1*. Statistical significance was assessed on unscaled data by one-way repeated-measures ANOVA for each cytokine. Treatment effect was significant for all cytokines after controlling for multiple comparisons using the two-stage linear step-up procedure of Benjamini, Krieger, and Yekutieli (FDR = 5%). Tukey's multiple comparisons test identified between-group differences. *p<0.05, **p<0.01, ***p<0.001, ****p<0.0001 vs. mock transfection; #p<0.05, ##p<0.01, ###p<0.001 vs. conventional saRNA. The mock transfection control is shared with panel (c). (**b**) Schematic of saRNA constructs designed for inhibiting inflammatory signaling. dsRNA-sensing pathway inhibitors and EGFP are expressed via an IRES. A second IRES expresses variable amounts of inflammatory signaling pathway inhibition: 'moxBFP' expresses moxBFP (control). 'srIκBα' expresses srIκBα (inhibiting NF-κB). 'srIκBα-Smad7-SOCS1' expresses srIκBα plus Smad7 (inhibiting TGF-β) and SOCS1 (inhibiting IFN). (**c**) Antiviral cytokine response quantified by multiplex bead-based immunoassay (n=6). srIκBα and srIκBα-Smad7-SOCS1 broadly suppressed cytokine secretion. Data normalization, visualization, and statistical analysis were performed as described in panel (a), but scaling was applied independently to account for the different constructs tested. Plots of unscaled data are presented in *Figure 4—figure supplement 2*. *p<0.05, **p<0.01, ***p<0.001, ****p<0.0001 vs. mock transfection; #p<0.05, ##p<0.01, ###p<0.001, ####p<0.0001 vs. moxBFP. The mock transfection control is shared with panel (a). Abbreviations: srIκBα, super repressor inhibitor of κBα; Smad7, mothers against decapentaplegic homolog 7; SOCS1, suppressor of cytokine signaling 1; IFN, interferon; CXCL, C-X-C motif chemokine

*Figure 4 continued on next page*

Figure 4 continued

ligand; TNF, tumor necrosis factor; MCP-1, monocyte chemoattractant protein-1; IL, interleukin; CCL5, chemokine ligand 5; GM-CSF, granulocyte-macrophage colony-stimulating factor.

The online version of this article includes the following source data and figure supplement(s) for figure 4:

**Source data 1.** Whole-plasmid sequencing results for plasmids depicted in *Figure 4b*.

**Figure supplement 1.** Detailed cytokine responses to saRNA constructs inhibiting dsRNA-sensing pathways.

**Figure supplement 1—source data 1.** Numerical data used to generate the plots in *Figure 4—figure supplement 1*.

**Figure supplement 2.** Detailed cytokine responses to saRNA constructs inhibiting inflammatory signaling pathways.

**Figure supplement 2—source data 1.** Numerical data used to generate the plots in *Figure 4—figure supplement 2*.

**Figure supplement 3.** Reduced antiviral gene expression and replicon activity observed with co-expression of inflammatory signaling inhibitors.

**Figure supplement 3—source data 1.** Numerical data used to generate the plots in *Figure 4—figure supplement 3*.

**Figure supplement 3—source data 2.** PDF file containing the original gel image for *Figure 4—figure supplement 3c*, with relevant bands labeled.

**Figure supplement 3—source data 3.** Unmodified/uncropped original gel image as shown in *Figure 4—figure supplement 3c*.

attenuates the cytokine response to saRNA. Further strategies are needed to reduce saRNA-induced cytokine secretion to levels acceptable for non-immunotherapeutic applications.

## Design of saRNA constructs for inhibiting multiple inflammatory signaling pathways

To further inhibit saRNA-induced cytokine release, we designed additional saRNA constructs that co-express cellular inhibitors of inflammatory signaling. In these constructs, dsRNA pathway inhibitors (vaccinia virus E3, Toscana virus NSs, Theiler's virus L*) and EGFP were expressed from a single IRES (*Figure 4b*). A second IRES was then used to express inflammatory signaling inhibitors.

Given that activation of nuclear factor-κB (NF-κB) is a hallmark of viral infection and a master regulator of cytokine and chemokine induction (*Mitchell and Carmody, 2018*; *Santoro et al., 2003*), we prioritized its inhibition using super repressor inhibitor of κBα (srIκBα). This dominant-active variant cannot be phosphorylated and thus resists ubiquitination and degradation, forming a stable cytoplasmic pool of IκBα that prevents NF-κB nuclear translocation and downstream signaling (*Brown et al., 1995*; *Lee et al., 2009*). We generated one construct expressing srIκBα alone (named 'srIκBα') and another expressing srIκBα in combination with Smad7 and suppressor of cytokine signaling 1 (SOCS1) using nonidentical 2A peptides (named 'srIκBα-Smad7-SOCS1'). Smad7 negatively regulates both transforming growth factor-β (TGF-β) and NF-κB signaling pathways (*Yan et al., 2009*), while SOCS1 inhibits type I, II, and III IFN pathways, as well as NF-κB signaling (*Dimitriou et al., 2008*; *Blumer et al., 2017*; *Sobah et al., 2021*). Additionally, a control construct ('moxBFP') was generated that lacks inflammatory signaling inhibitors and instead expresses a blue fluorescent protein.

## Inhibiting NF-κB signaling broadly attenuates saRNA-induced antiviral cytokine secretion

Transfection with the moxBFP construct, which lacks inflammatory signaling inhibitors, led to a significant increase in the secretion of all measured cytokines (*Figure 4c*, *Figure 4—figure supplement 2*). In contrast, co-expression of srIκBα broadly reduced cytokine levels, significantly reducing the secretion of all cytokines except TNF, IFN-α, and interleukin (IL)-6. The srIκBα-Smad7-SOCS1 construct did not further enhance suppression relative to srIκBα alone, with the exception of IL-6, which was significantly reduced only by srIκBα-Smad7-SOCS1. These results indicate that NF-κB is a key driver of saRNA-induced cytokine production, and that co-expression of srIκBα is an effective strategy for attenuating this response.

## Reduced antiviral gene expression and replicon activity observed with co-expression of inflammatory signaling inhibitors

To evaluate whether co-expression of inflammatory signaling inhibitors could attenuate saRNA-induced induction of antiviral transcripts, we performed an RT-qPCR array on mock-transfected cells and cells transfected with conventional saRNA, E3, or srIκBα-Smad7-SOCS1 (*Figure 4—figure*

*supplement 3a*). Transfection with conventional saRNA led to marked upregulation of several antiviral and proinflammatory mRNA transcripts, including *Adar*, *Isg20*, *Rigi*, *Ifih1*, *Tlr3*, *Eif2ak2*, *Zc3hav1*, and *Il6*. In contrast, both the E3 and srIκBα-Smad7-SOCS1 constructs attenuated the induction of these transcripts, with srIκBα-Smad7-SOCS1 exhibiting broader suppression across the panel.

Because all saRNA constructs in this study encode EGFP, we quantified its transcript levels as a proxy for replicon amplification, although we note that this interpretation assumes a consistent genomic-to-subgenomic transcript ratio across constructs. FLS transfected with srIκBα-Smad7-SOCS1 exhibited significantly lower *EGFP* transcript levels than those transfected with conventional saRNA or E3 (*Figure 4—figure supplement 3b*). This apparent reduction in replicon activity is likely multifactorial. For one, physical characteristics of the transcripts contribute: the srIκBα-Smad7-SOCS1 construct is ~40% longer, resulting in a lower molar input when transfected at equal RNA mass. Its greater length is also expected to increase replication time, potentially altering replication dynamics. For another, the moxBFP, srIκBα, and srIκBα-Smad7-SOCS1 constructs share a common truncated byproduct (*Figure 4—figure supplement 3c and d*). This byproduct is expected to be replication-incompetent, as it lacks the conserved sequence element in the 3′ untranslated region necessary for negative-strand synthesis (*Hyde et al., 2015*), thus further reducing the effective molar input of functional saRNA. And finally, biological effects of the co-expressed proteins may also limit replicon

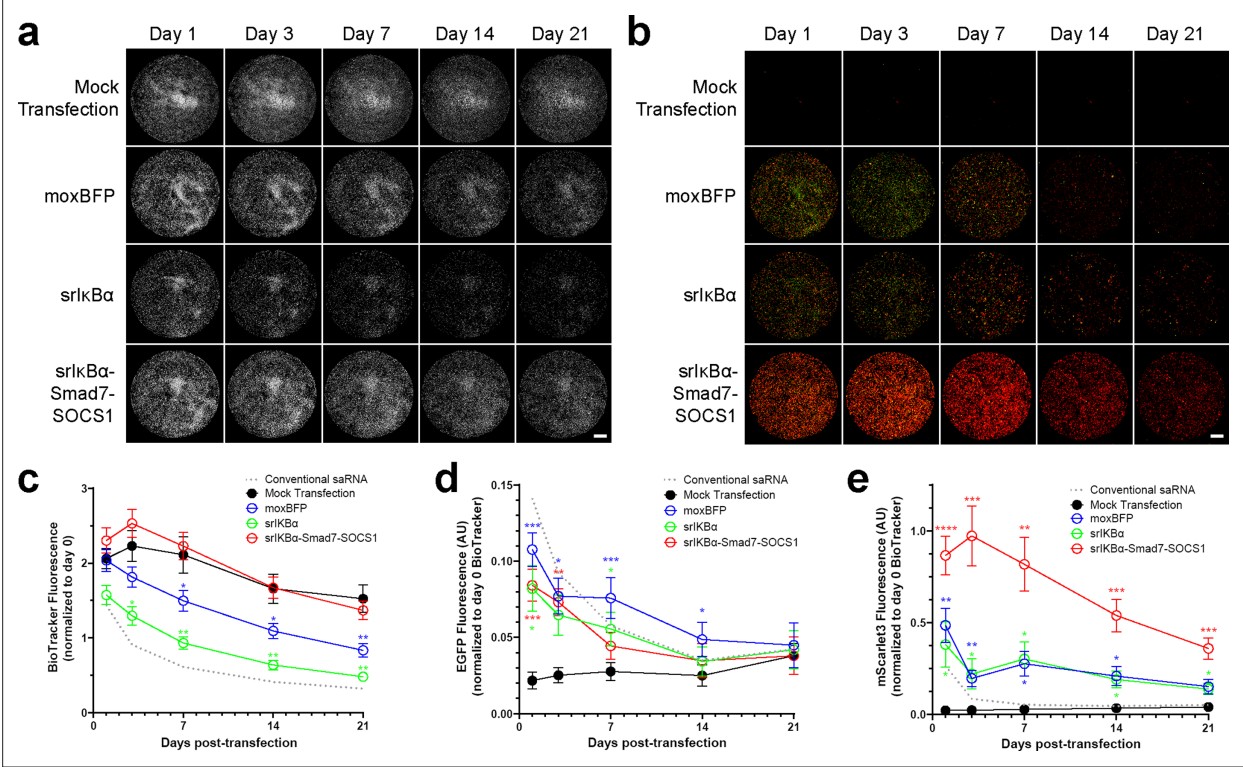

**Figure 5.** Differential effects of srIκBα and srIκBα-Smad7-SOCS1 on cell number and transgene expression. (**a**) Representative images of microplate wells showing BioTracker. (**b**) Representative composite images of microplate wells showing EGFP (green) and mScarlet3 (red). (**c**) Longitudinal quantification of BioTracker (n=11). srIκBα reduces signal, an effect not observed with srIκBα-Smad7-SOCS1.(**d**) Longitudinal quantification of EGFP (n=11). Expression was low across constructs. (**e**) Longitudinal quantification of mScarlet3 (n=11). srIκBα-Smad7-SOCS1 produced 2–3 times greater fluorescence than other constructs. For panels (a–b): Scale bar = 5 mm. For panels (c–e): Data are normalized to starting cell number (pre-transfection BioTracker signal). A dotted line shows the response to conventional saRNA (for reference). Statistical significance of treatment effects at each time point compared to mock transfection was determined by two-way RM ANOVA with Greenhouse-Geisser correction and Dunnett's multiple comparisons test. *p<0.05, **p<0.01, ***p<0.001, and ****p<0.0001. Data are presented as mean ± SEM. The mock transfection control data used in this figure is also presented in *Figure 1d–f*.

The online version of this article includes the following source data and figure supplement(s) for figure 5:

**Source data 1.** Numerical data used to generate the plots in *Figure 5*.

**Figure supplement 1.** srIκBα induces cell loss, an effect not observed with srIκBα-Smad7-SOCS1.

**Figure supplement 1—source data 1.** Numerical data used to generate the plots in *Figure 5—figure supplement 1*.

activity. For example, inhibition of RNase L by L* could increase competition for cellular resources and thereby reduce RdRp production.

## Co-expression of Smad7 and SOCS1 rescues srIκBα-induced cell loss and enhances transgene expression

We then examined the impact of inhibiting inflammatory signaling on cell number and transgene expression (*Figure 5a and b*). Interestingly, the srIκBα construct caused both immediate and sustained reductions in cell number, as quantified by BioTracker signal, whereas the srIκBα-Smad7-SOCS1 construct maintained cell number comparable to mock transfection throughout the experiment (*Figure 5c*). AUC analysis of the BioTracker data confirmed that srIκBα-Smad7-SOCS1 significantly mitigated the srIκBα-induced reduction in integrated BioTracker signal (*Figure 5—figure supplement 1a*). Additional experiments using CellTag and calcein AM staining corroborated these findings, showing that srIκBα negatively impacted both cell number (*Figure 5—figure supplement 1b*) and viability (*Figure 5—figure supplement 1c*), effects that were prevented by co-expression of Smad7 and SOCS1.

All three constructs produced low levels of EGFP expression (*Figure 5d*), which was expected given that EGFP is the fourth cistron in a 2A-linked polyprotein (*Figure 4b*). Previous studies have established that protein expression decreases significantly for downstream positions in such 2A-linked configurations (*Liu et al., 2017a*). Notably, the srIκBα-Smad7-SOCS1 construct produced two- to threefold higher mScarlet3 fluorescence than either the moxBFP or srIκBα constructs (*Figure 5e*), demonstrating that the srIκBα-Smad7-SOCS1 construct supports robust transgene expression despite its multi-cistronic configuration.

## Smad7 and SOCS1 co-expression prevents srIκBα-induced alterations in translational control

We next investigated the molecular mechanisms by which srIκBα and srIκBα-Smad7-SOCS1 modulate translational control. Interestingly, srIκBα expression led to reductions in eIF2α phosphorylation, a change that was prevented by co-expression of Smad7 and SOCS1 (*Figure 6a*). While neither construct affected total eIF2α levels or eIF4E phosphorylation compared to the moxBFP construct (*Figure 6b and c*), srIκBα caused a significant reduction in total eIF4E levels (*Figure 6d*). Given the critical role of eIF4E in cap-dependent translation, its depletion likely contributes to the poor transgene expression observed with srIκBα (*Figure 5e*). Importantly, Smad7 and SOCS1 co-expression preserved eIF4E levels, counteracting the disruption of cap-dependent translation caused by srIκBα.

## Prolonged expression of moxBFP does not activate FLS, and srIκBα co-expression reduces basal activation

Fibroblast activation protein-α (FAP-α) is a serine protease expressed on the surface of FLS that contributes to extracellular matrix degradation and tissue remodeling (*Bauer et al., 2006*; *Zhang et al., 2019*). Its expression is minimal in normal adult FLS but increases significantly following inflammatory activation, such as in rheumatoid arthritis and osteoarthritis (*Croft et al., 2019*; *Fan et al., 2023*). Since viral infections can induce fibroblast activation (*Boyd et al., 2020*; *Krausgruber et al., 2020*), we hypothesized that prolonged saRNA expression might similarly trigger FLS activation. To test this, we measured FAP-α levels using an in-cell western assay 11 days post-transfection. FLS transfected with moxBFP showed FAP-α levels comparable to mock-transfected controls, whereas cells transfected with srIκBα and srIκBα-Smad7-SOCS1 exhibited significantly lower FAP-α levels (*Figure 7a and b*). These findings suggest that prolonged saRNA expression from constructs encoding inhibitors of dsRNA-sensing pathways does not overtly trigger fibroblast activation, whereas co-expression of inhibitors of inflammatory signaling may reduce basal activation.

## Reversible and irreversible external control of immune-evasive saRNA

The srIκBα-Smad7-SOCS1 construct represents a promising immune-evasive saRNA platform for gene therapy and other non-immunotherapeutic applications. It includes the E3-NSs-L* polyprotein, which protects against saRNA-induced cytotoxicity (*Figure 1f*; *Figure 3—figure supplement 1a and b*; *Figure 2b and d*), translation shutdown (*Figure 1e*; *Figure 3a*), and host mRNA degradation (*Figure 3—figure supplement 1b and c*). In parallel, co-expression of srIκBα attenuates

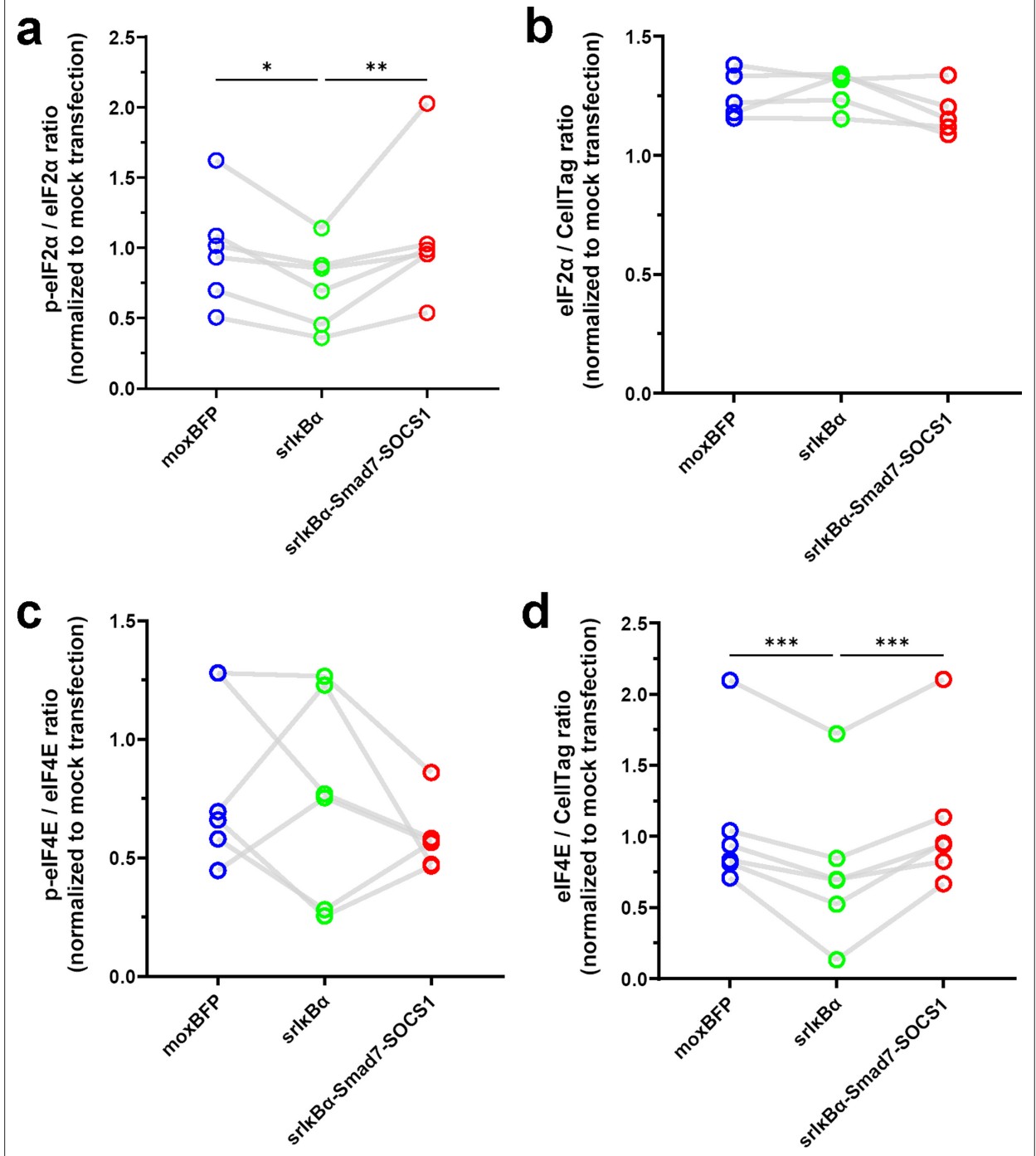

**Figure 6.** srIκBα reduces eIF2α phosphorylation and total eIF4E levels, effects not observed with srIκBα-Smad7-SOCS1. (**a**) Phosphorylated eIF2α levels examined by in-cell western, normalized to total eIF2α levels (n=6). Phosphorylation was reduced by srIκBα but unchanged by srIκBα-Smad7-SOCS1. (**b**) Total eIF2α levels examined by in-cell western, normalized to CellTag (n=5). No significant differences were revealed ($F_{(2,8)}$ = 3.683, p=0.0735). (**c**) Phosphorylated eIF4E levels examined by in-cell western, normalized to total eIF4E levels (n=6). No significant differences were revealed ($F_{(2,10)}$ = 1.336, p=0.3059). (**d**) Total eIF4E levels examined by in-cell western, normalized to CellTag (n=6). srIκBα reduced total eIF4E, while srIκBα-Smad7-SOCS1 had no effect. For all panels: Data are presented as fold change relative to mock transfection. Statistical significance was determined by one-way RM ANOVA and Holm-Šídák's multiple comparisons test to compare all groups. *p<0.05, **p<0.01, and ***p<0.001. Connecting lines indicate responses from the same biological replicate. Mock transfection data used for normalization are the same as in *Figure 3*.

The online version of this article includes the following source data for figure 6:

**Source data 1.** Numerical data used to generate the plots in *Figure 6*.

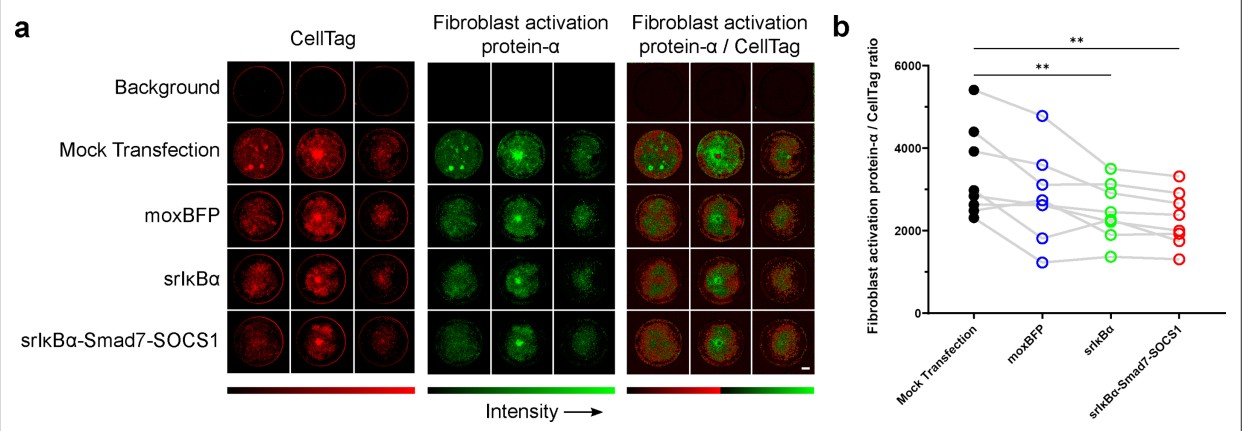

**Figure 7.** Long-term expression of saRNA constructs inhibiting inflammatory signaling reduces basal fibroblast activation factor-α (FAP-α) levels. (**a**) Representative in-cell western images showing FAP-α, a marker of fibroblast activation, on day 11 post-transfection. Columns show different biological replicates, and rows show different treatments. The tiled composite on the right shows FAP-α signal normalized to CellTag signal (FAP-α/CellTag). (**b**) Quantification of FAP-α, normalized to CellTag (n=8). Both srIκBα and srIκBα-Smad7-SOCS1 significantly reduce FAP-α levels versus mock transfection, while moxBFP shows no significant difference. Statistical significance was determined by one-way RM ANOVA with Greenhouse-Geisser correction and Dunnett's multiple comparisons test to compare groups to mock transfection. Connecting lines indicate responses from the same biological replicate. **p<0.01.

The online version of this article includes the following source data for figure 7:

**Source data 1.** Numerical data used to generate the plots in *Figure 7*.

saRNA-induced cytokine responses (*Figure 4c*), while Smad7 and SOCS1 mitigate srIκBα-associated reductions in cell viability (*Figure 5c*; *Figure 5—figure supplement 1a–c*) and eIF4E levels (*Figure 6d*). These combined features enable sustained cap-dependent transgene expression (*Figure 5e*) without triggering fibroblast activation (*Figure 7b*).

To improve safety and flexibility, we next explored whether transgene expression could be externally regulated using a small-molecule antiviral. ML336 is a potent inhibitor of the VEEV RdRp used by the saRNA replicon (*Chung et al., 2010*). To assess its ability to modulate transgene expression, we conducted a concentration–response assay by adding ML336 to the culture medium at the time of transfection. ML336 suppressed mScarlet3 expression in a concentration-dependent manner with an $IC_{50}$ of 8.5 nM (*Figure 8a*), demonstrating tunable control of saRNA-driven transgene expression.

We then investigated whether this inhibition was reversible or tied to permanent loss of replicon activity. In a washout experiment, cultures from the initial concentration–response assay were washed 2 days post-transfection, and mScarlet3 fluorescence was quantified the following day. Recovery of mScarlet3 expression occurred at intermediate ML336 concentrations (10 nM–1 µM), peaking at 100 nM (*Figure 8b*), a range that likely reflects conditions where genomic RNA replication persists at low levels but subgenomic RNA synthesis is curtailed. In contrast, recovery did not occur at concentrations above 1 µM. This suggests that inhibition of transgene expression is reversible at intermediate ML336 concentrations, while sustained replicase inhibition at higher concentrations leads to irreversible replicon loss and permanent termination of transgene expression.

To determine whether ML336 could inhibit an already active replicon, we allowed replicon activity to proceed for 1 day before administering 1 µM ML336 or vehicle control. Cultures were then monitored over a 12-day period (*Figure 8c*). Continued ML336 treatment led to a sustained decline in both EGFP (*Figure 8e*) and mScarlet3 expression (*Figure 8f*), confirming effective external control of an established immune-evasive replicon.

Importantly, prolonged ML336 exposure did not compromise cell viability relative to mock-transfected controls (*Figure 8d and g*). By contrast, vehicle-treated cultures—where saRNA replication remained unchecked—showed significantly reduced calcein AM signal. This difference may reflect altered cell proliferation over the course of the experiment, consistent with our earlier finding that saRNA reduces eIF4E phosphorylation (*Figure 3c*), a key regulator of cell growth and proliferation (*Muta et al., 2011*).

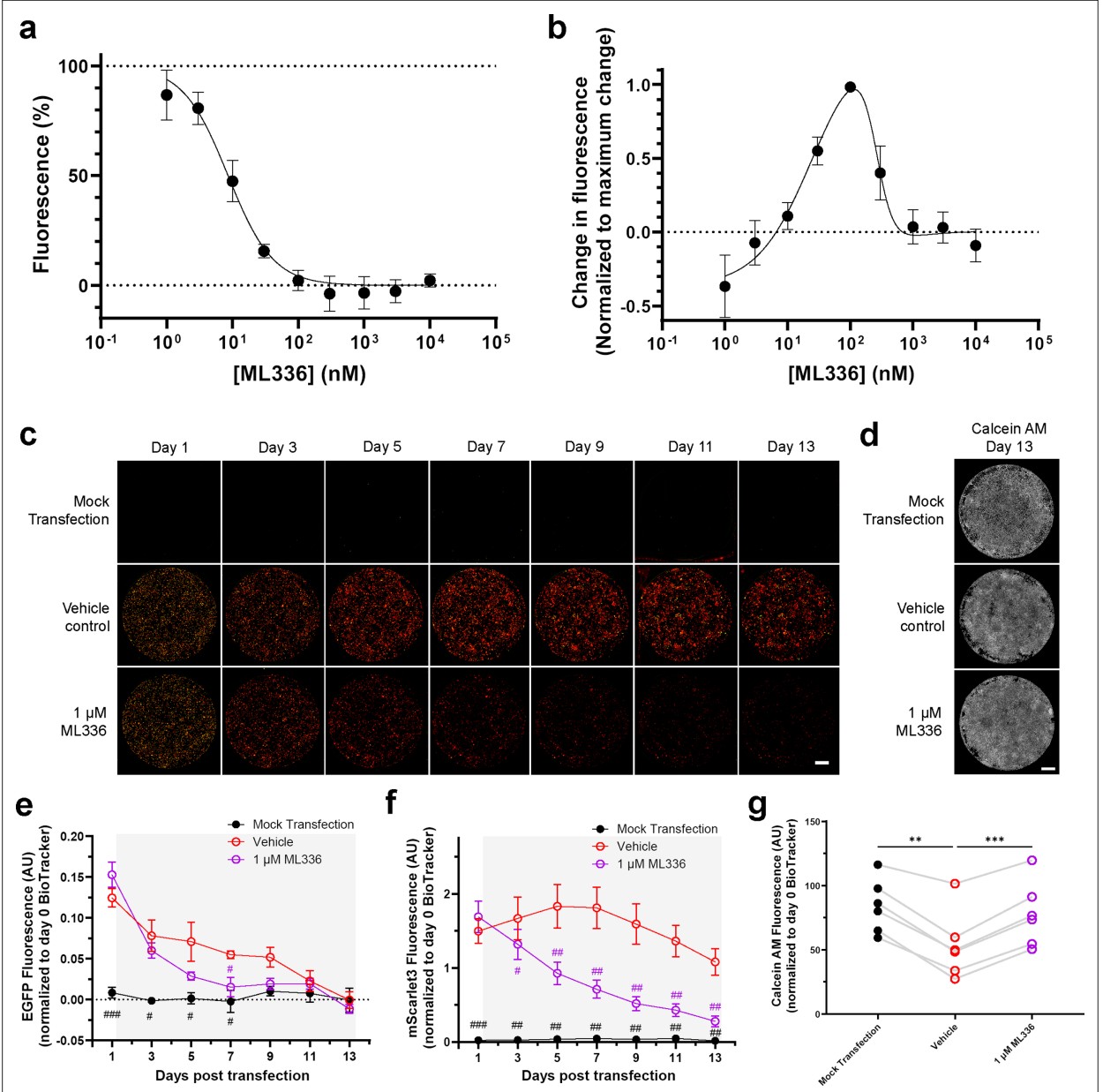

**Figure 8.** Reversible and irreversible control of saRNA replicon activity with ML336. Panels (**a**–**b**) show the effect of ML336, a small-molecule replicase inhibitor, applied at the time of transfection with srIκBα-Smad7-SOCS1. (**a**) ML336 concentration–response curve for inhibiting mScarlet3 expression, measured 2 days post-transfection (n=5). IC$_{50}$ = 8.5 nM (95% CI: 6.1–11.9 nM). Data are normalized to vehicle-treated controls and fit to a variable-slope sigmoidal curve. (**b**) Reversibility of ML336-mediated inhibition of mScarlet3 expression determined by measuring fluorescence recovery, one day after compound washout from cultures in panel (a). mScarlet3 inhibition is reversible at intermediate concentrations (10 nM–1 μM) but irreversible at high concentrations (>1 μM). Data are normalized to each biological replicate's maximum recovery and fit to a bell-shaped concentration–response curve. Panels (**c**–**g**) show the effect of ML336 on an established srIκBα-Smad7-SOCS1 replicon. FLS were continously treated (indicated by shading) with vehicle or 1 μM ML336, starting 1 day post-transfection. (**c**) Representative composite images of microplate wells showing EGFP (green) and mScarlet3 (red). Scale bar = 5 mm. (**d**) Representative images of microplate wells showing calcein AM at end of time course. Scale bar = 5 mm. (**e**) Quantification of EGFP (n=6). By day 7, EGFP signal is reduced by ML336. (**f**) Quantification of mScarlet3 (n=6). By day 3, mScarlet3 signal is reduced by ML336. (**g**) Quantification of calcein AM (n=6). Vehicle, but not ML336 treatment, reduced calcein AM signal relative to mock transfection. Statistical significance was assessed by one-way RM ANOVA with Greenhouse-Geisser correction and Tukey's multiple comparisons test. **p<0.01, ***p<0.001. All fluorescence data were normalized to pre-transfection cell number indicated by BioTracker signal. Connecting lines indicate responses from the same biological replicate. For panels (e–f): Statistical significance relative to vehicle-treated cells was determined by two-way RM ANOVA with Greenhouse-Geisser correction and Dunnett's multiple comparisons test. #p<0.05, ##p<0.01, ###p<0.001. Data are presented as mean ± SEM.

The online version of this article includes the following source data for figure 8:

**Source data 1.** Numerical data used to generate the plots in **Figure 8**.

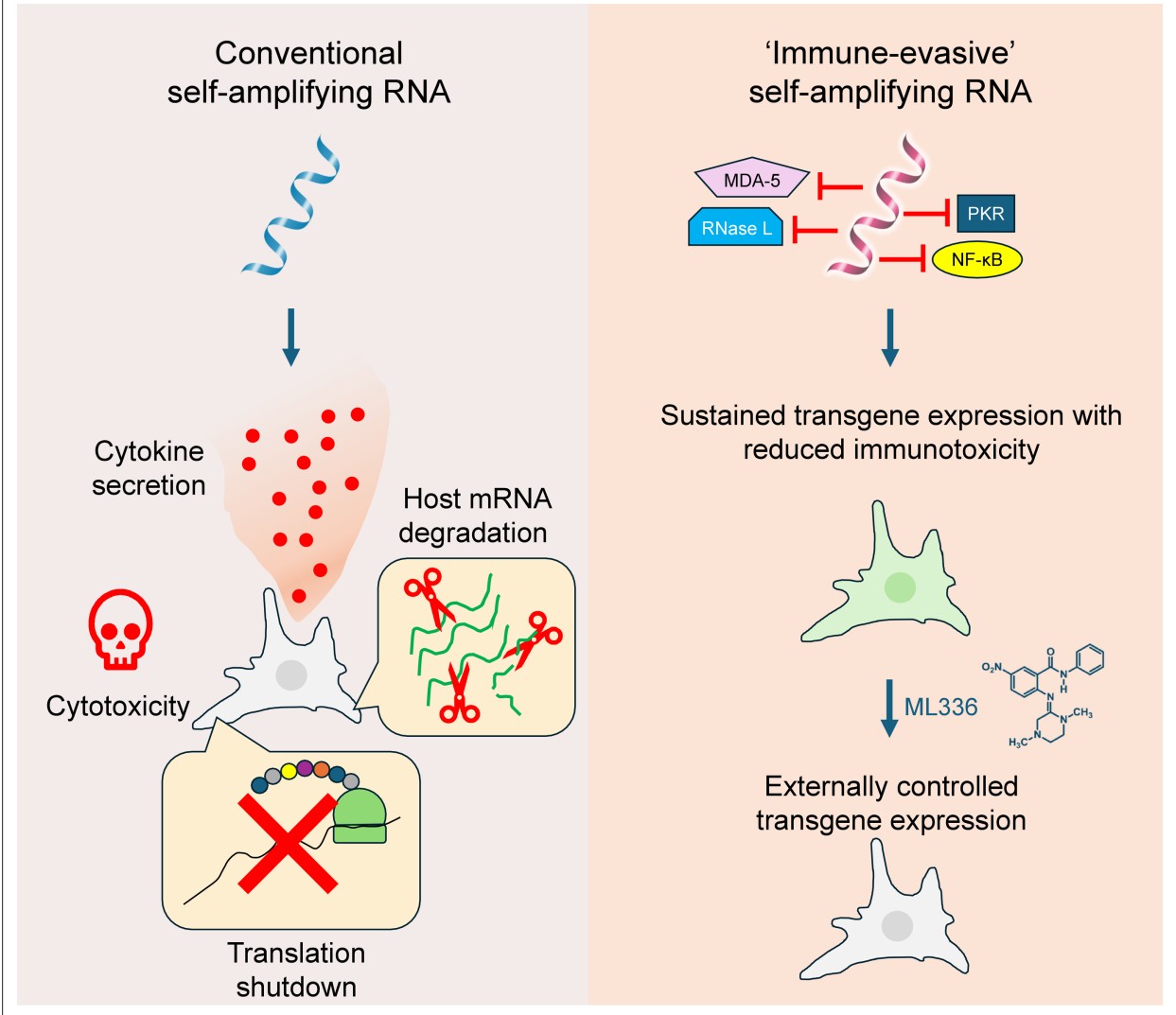

**Figure 9.** Diagrammatic comparison of conventional and immune-evasive self-amplifying RNA (saRNA). Conventional saRNA activates multiple innate immune responses upon replication, leading to translation shutdown, host mRNA degradation, cytotoxicity, and cytokine secretion. These immunotoxic effects restrict its therapeutic utility to vaccine applications and often necessitate co-administration of exogenous immunosuppressants to enable effective transgene expression. In contrast, immune-evasive saRNA co-expresses inhibitors of key dsRNA-sensing and inflammatory signaling pathways via cap-independent translation, intrinsically suppressing these immune responses. This enables sustained transgene expression while reducing immunotoxicity and eliminating the need for exogenous immunosuppressive agents. Transgene expression can be externally regulated or terminated using small-molecule inhibitors of the saRNA replicase, such as ML336, providing control over the duration of therapy and a means to mitigate potential adverse effects.

## Discussion

Although saRNA is known to disrupt translational control, the precise mechanisms and downstream effects remain incompletely characterized. saRNA-induced translation shutdown is thought to be mediated by phosphorylation of eIF2α (*Beissert et al., 2017*; *Frolova et al., 2023*; *Minnaert et al., 2021*), following the activation of PKR in response to dsRNA replicative intermediates (*Gorchakov et al., 2004*; *Beissert et al., 2017*). Our findings support this model and further show that saRNA reduces eIF4E phosphorylation, a phenomenon previously observed with alphavirus replicons, though its mechanism remains unclear (*Berglund et al., 2007*; *Fros and Pijlman, 2016*). Reduced eIF4E phosphorylation suppresses cap-dependent translation initiation (*Scheper and Proud, 2002*), thereby favoring cap-independent translation (*Dyer et al., 2003*). Together, eIF2α phosphorylation and reduced eIF4E activity likely explain why IRES-driven transgenes achieve higher expression than their

cap-dependent counterparts when using conventional saRNA (*Erasmus et al., 2020*; *Chang et al., 2022*).

In addition to disrupting translation initiation, saRNA replication induces host translation shutoff through activation of RNase L. Many viruses exploit host shutoff—the global suppression of host gene expression—as a strategy to redirect cellular resources toward their replication (*Gaucherand and Gaglia, 2022*). Our study suggests that saRNA also co-opts this mechanism. dsRNA intermediates generated during saRNA replication activate the OAS/RNase L pathway. RNase L activation then degrades host mRNAs without impairing the translation machinery (*Donovan et al., 2017*). saRNA transcripts, on the other hand, are largely protected from degradation due to their localization within viral replication organelles known as spherules (*Pietilä et al., 2017*; *Minnaert et al., 2021*). This favors their translation over degraded host transcripts.

The degradation of host transcripts not only adversely affects the host translational landscape, but also contributes to cytotoxicity. The cleaving of actively translated mRNAs leads to ribosome stalling at 3' ends and ribosome collisions (*Guydosh et al., 2017*; *Xi et al., 2024*), activation of the ribotoxic stress response (*Snieckute et al., 2022*), and ultimately, apoptosis (*Li et al., 2004*; *Xi et al., 2024*). Similarly, PKR likely contributes to saRNA-induced cytotoxicity. Previous studies have shown that microinjected dsRNA induces apoptosis in a PKR-dependent manner in HeLa cells, with RIG-I and MDA-5 being dispensable (*Zuo et al., 2022*). Many additional studies further support PKR as a central mediator of apoptosis in response to cytosolic dsRNA (*Xu et al., 2018*; *Gil and Esteban, 2000*; *Zhang and Samuel, 2007*). Notably, apotosis mediated by PKR can occur independently of eIF2α phosphorylation (*von Roretz and Gallouzi, 2010*), suggesting that inhibiting PKR may not fully prevent its cytotoxic actions.

The cytotoxic effects triggered by dsRNA-sensing pathways are highlighted by the E3 construct, which alleviates translation shutdown but does not prevent RNase L activation or PKR upregulation. While RNase L activity enables high transgene expression, it comes at a cost: long-term cytotoxicity. This dual-edged effect may make the E3 construct particularly well suited for immunotherapeutic applications such as vaccines or oncolytic-like therapies, where robust transgene expression and immunogenic cell death are advantageous (*Leitner et al., 2004*; *Wu et al., 2023*). In contrast, applications requiring both sustained transgene expression and cell viability demand strategies that address not only saRNA-induced alterations in translational control but also cytotoxicity. The E3-NSs-L* polyprotein, which combines broad inhibition of dsRNA-sensing pathways with targeted suppression of RNase L and ubiquitin-ligase–mediated degradation of PKR, effectively protects against both mechanisms, making it an essential component of immune-evasive saRNA.

However, inhibiting dsRNA-sensing pathways was not effective at preventing saRNA-induced cytokine responses. This agrees with prior reports showing that dsRNA activates NF-κB even in the absence of PKR and RNase L (*Iordanov et al., 2001*), suggesting that alternative pathways contribute to NF-κB activation. One such mechanism may involve eIF4E, whose activity is governed by its phosphorylation status and serves as a rate-limiting step in cap-dependent protein synthesis (*Mars et al., 2024*). In this study, we found that saRNA induces a reduction in eIF4E phosphorylation. Reduced eIF4E phosphorylation has been shown to decrease the levels of the short-lived IκBα protein (*Herdy et al., 2012*), an inhibitor of NF-κB that sequesters it in the cytoplasm and prevents nuclear translocation. Decreased IκBα levels thus allow NF-κB to translocate to the nucleus, initiating cytokine expression and inflammatory responses (*Liu et al., 2017b*). This mechanistic insight provided a clear rationale for the next step in our design of the immune-evasive saRNA platform: inhibition of the NF-κB pathway with srIκBα.

Importantly, while NF-κB is best known for promoting inflammatory cytokine production, it also supports cell survival by inducing anti-apoptotic genes, though its effects vary by tissue type and biological context (*Luo et al., 2005*). In FLS, overexpression of srIκBα reduces cytokine production but increases sensitivity to TNF-induced apoptosis (*Miagkov et al., 1998*; *Zhang et al., 2000*), consistent with our findings that saRNA encoding srIκBα lowered both cytokine responses and cell viability. This phenomenon is not unique to FLS; NF-κB inhibition with srIκBα also renders liver hepatocytes more susceptible to pro-apoptotic stimuli (*Xu et al., 1998*), raising safety concerns given that RNA delivery systems such as lipid nanoparticles (LNPs) tend to accumulate in the liver (*Sago et al., 2019*). Therefore, when designing immune-evasive saRNA constructs, the use of srIκBα to suppress cytokine responses may require co-expression of anti-apoptotic proteins to offset the loss of NF-κB-mediated

survival signaling. In our case, Smad7 and SOCS1 restored FLS viability when co-expressing srIκBα. While we did not assess the individual contributions of each protein, both are known to have anti-apoptotic properties in certain contexts. Smad7 can inhibit TGF-β-induced apoptosis (*Patil et al., 2000*; *Yamamura et al., 2000*), while SOCS1 can suppress apoptosis triggered by IFN-α, IFN-β (*Zitzmann et al., 2007*), IFN-γ (*Mann-Chandler et al., 2005*), and TNF-α (*Yan et al., 2008*; *Madonna et al., 2012*; *Kimura et al., 2004*). These anti-apoptotic effects likely enable Smad7 and SOCS1 to mitigate srIκBα-induced reductions in FLS cell viability.

We also found that co-expression of Smad7 and SOCS1 significantly enhanced cap-dependent transgene expression. Although we did not distinguish the individual contributions of each protein, this effect is likely mediated by SOCS1. This hypothesis is supported by previous studies showing that ruxolitinib—a small-molecule inhibitor of Janus kinase (JAK) 1 and JAK2 (*Goker Bagca and Biray Avci, 2020*)—increases saRNA transgene expression when added to saRNA polyplexes (*Blakney et al., 2021b*). As SOCS1 is an endogenous inhibitor of JAK1 and JAK2 (*Liau et al., 2018*), it offers a genetically encoded alternative to the small-molecule approach of ruxolitinib for enhancing saRNA-driven transgene expression. The mechanisms underlying this enhancement are unclear, and our study found no evidence implicating eIF2α or eIF4E in this effect. Potentially, it may arise from non-canonical translational control mechanisms, or from increased subgenomic RNA levels. Since our study did not examine amplification dynamics, future investigations will clarify these possibilities.

Although our study focused on engineering immune-evasive saRNA to support sustained transgene expression in non-immunotherapeutic settings, the platform may also offer benefits for vaccine development. Both preclinical and clinical data indicate that saRNA vaccines often trigger an intense innate immune response, which can hinder antigen expression (*Dominguez et al., 2023*; *Blakney et al., 2021b*; *Beissert et al., 2017*). Additionally, saRNA vaccination is associated with pronounced inflammation (*Bathula et al., 2024*; *Tregoning et al., 2023*), contributing to commonly reported adverse events such as pain, headache, tenderness, arthralgia, fever, and chills (*Oda et al., 2024*; *Saraf et al., 2024*). By dampening the self-adjuvanting (immunostimulatory) effects of saRNA replication, the immune-evasive saRNA platform could enhance antigen production while reducing reactogenicity—offering a promising strategy for next-generation saRNA vaccines.

For applications requiring sustained therapeutic gene expression, a key advantage of the immune-evasive saRNA platform over traditional viral vectors is its potential for precise external control. While viral vectors can support long-term transgene expression, they generally lack external shut-off mechanisms or straightforward means for vector elimination—often relying on more extreme strategies such as suicide genes to ensure safety (*Gargett and Brown, 2014*; *Greco et al., 2015*). In contrast, saRNA replicons can be modulated using small-molecule inhibitors targeting the RdRp. Here, we demonstrate that ML336, a potent and well-tolerated VEEV RdRp inhibitor (*Chung et al., 2010*; *Schroeder et al., 2014*; *Skidmore et al., 2020*), enables concentration-dependent tuning of immune-evasive saRNA-driven transgene expression, including reversible inhibition at intermediate concentrations and irreversible termination at higher concentrations. Critically, ML336 remains effective even when administered after replicon activity is established and does not compromise cell viability. These features collectively position RdRp inhibitors as a practical and versatile means to externally control immune-evasive saRNA—either to transiently suppress transgene expression or to permanently eliminate the replicon once therapeutic goals are achieved.

Another important advantage of the immune-evasive saRNA platform is that it does not require exogenous immunosuppressants, which are commonly needed in non-immunotherapeutic applications of conventional saRNA (*Vanluchene et al., 2024*; *Zhong et al., 2021*; *Yoshioka et al., 2013*; *Erasmus et al., 2020*; *Kim et al., 2017*). By restricting immune suppression to transfected cells, this localized strategy has the potential to reduce treatment complexity and mitigate off-target risks associated with systemic immunosuppression. However, the downside of this all-in-one design is that it requires manufacturing long RNA molecules. Producing long RNA with high yield and integrity presents technical challenges (*Rosa et al., 2021*). Larger RNAs are more prone to secondary structure formation, increasing the likelihood of abortive transcription during in vitro synthesis (*Beissert et al., 2020*). Overcoming this limitation may be achievable through optimization of in vitro transcription conditions (*Samnuan et al., 2022*), or by applying computational strategies. For instance, deep-learning models developed for structural biology (*Shen et al., 2024*; *Kagaya et al., 2025*) could be

leveraged using transfer learning (*Ouellet et al., 2023*; *Ferguson et al., 2025*) to guide the rational design of transcripts with minimal secondary structure and improved full-length yield.

Further future directions include characterizing the in vivo effectiveness of the immune-evasive saRNA platform. In a living organism, saRNA-transfected cells must contend with two forms of cytotoxicity. They face not only intrinsic cytotoxicity related to saRNA replication (addressed in this study), but also extrinsic cytotoxicity mediated by immune cells. Though our study demonstrates that immune-evasive saRNA dramatically reduces cytokine release compared to conventional saRNA, the residual levels may still be sufficient to recruit immune effector cells. Furthermore, the co-expression of several viral proteins, along with the VEEV RdRp, increases the likelihood that viral peptides will be presented on major histocompatibility complex class I molecules. This, in turn, is expected to promote recognition and elimination of saRNA-transfected cells by cytotoxic T lymphocytes, thereby limiting therapeutic durability. Elucidating the spatiotemporal dynamics of transfected cell fate and their interactions with immune cells with techniques like multiphoton microscopy (*Lim and Ruthazer, 2021*) will be critical for addressing this challenge and testing mitigation strategies. Overcoming both intrinsic and extrinsic cytotoxicity will be essential for achieving long-term persistence of saRNA-transfected cells and realizing the platform's full therapeutic potential.

Beyond elimination of saRNA-transfected cells by immune effector cells, another key consideration for future translation is determining whether the immune-evasive saRNA platform exhibits a distinct in vivo expression profile compared to conventional saRNA. Although both mRNA-LNPs and saRNA-LNPs share similar biodistribution—primarily accumulating in the liver—conventional saRNA typically fails to drive hepatic expression and instead induces transgene expression in the spleen, lungs, and muscle (*Kimura et al., 2023*; *Bathula et al., 2024*). One possibility for this discrepancy is that it arises from cell type-dependent differences in the detection or downstream response to cytosolic dsRNA intermediates generated by saRNA. As we observed in FLS, replication of conventional saRNA triggers translation shut down and cytotoxicity. These effects, mediated by cytosolic dsRNA-sensing pathways, ultimately limit transgene expression. Potentially, the same mechanisms may restrict saRNA-driven expression in other cell types. If so, by suppressing dsRNA-sensing pathways, the immune-evasive saRNA platform may broaden the range of cell types permissive to saRNA-driven expression. Further investigation into the in vivo expression profile and cell-type specificity will be essential, not only to harness this potential but also to identify any unanticipated effects before clinical translation.

Our findings present a fully saRNA-based approach that addresses a key barrier to the use of saRNA in non-immunotherapeutic applications: innate immune activation triggered by replicon activity. By encoding proteins that inhibit both dsRNA-sensing pathways and NF-κB signaling directly within the saRNA—expressed via cap-independent translation—we achieved sustained transgene expression with reduced immunotoxicity, eliminating the need for exogenous immunosuppressants (*Figure 9*). Furthermore, the use of ML336 enables on-demand regulation or termination of transgene expression, enhancing both the safety and versatility of this approach. Though our study was a proof-of-concept performed in vitro using primary mouse cells, given the established success of saRNA in preclinical models (*Gong et al., 2024*; *Blakney et al., 2021b*; *Beissert et al., 2017*; *Erasmus et al., 2020*) and approved human vaccines (*Ho et al., 2024*; *Saraf et al., 2024*), we anticipate that this approach will be readily translatable to future in vivo studies. Ultimately, continued refinement of the immune-evasive saRNA platform and delivery strategies will be crucial for realizing its potential to bridge the therapeutic gap between the transience of mRNA and the permanence of viral vectors.

## Materials and methods

### Key resources table

| Reagent type (species) or resource | Designation | Source or reference | Identifiers | Additional information |
|---|---|---|---|---|
| Recombinant DNA reagent | TagGFP2 Simplicon Plasmid (E3L) | Merck-Millipore | SCR725 | |
| Recombinant DNA reagent | pIRES2-EGFP | Clontech | 6029-1 | |
| Recombinant DNA reagent | pDx_mScarlet3 | Addgene | 189754 (RRID:Addgene_189754) | A gift from Dorus Gadella |
| Gene (Engineered fluorescent protein) | moxBFP | FPbase | SSTDU | Amino acid sequence |

*Continued on next page*

*Continued*

| Reagent type (species) or resource | Designation | Source or reference | Identifiers | Additional information |
|---|---|---|---|---|
| Gene (Engineered fluorescent protein) | mEGFP | FPbase | QKFJN | Amino acid sequence |
| Gene (*Orthopoxvirus vaccinia*—Strain: Western Reserve) | Vaccinia Virus E3 protein | UniProt | P21605-1 [1991-05-01 v1] | Amino acid sequence |
| Gene (*Phlebovirus toscanaense*) | Toscana virus NSs protein | UniProt | P21699 [1991-05-01 v1] | Amino acid sequence |
| Gene (*Cardiovirus theileri*—Strain: DA) | Theiler's virus L* protein | UniProt | P0DJX4 [2013-07-24 v1] | Amino acid sequence |
| Gene (*Alphapermutotetravirus thoseae*) | T2A | UniProt | Q9YK87 (139–156) [1999-05-01 v1] | Amino acid sequence |
| Gene (*Teschovirus asilesi*) | P2A | UniProt | A0A077CZU0 (977–995) [2014-10-29 v1] | Amino acid sequence |
| Gene (*Aphthovirus burrowsi*) | E2A | UniProt | K9MZ26 (991–1010) [2013-03-06 v1] | Amino acid sequence |
| Gene (*Mus musculus*) | IκBα | UniProt | Q9Z1E3 [2004-10-25 v2] | Amino acid sequence |
| Gene (Engineered mouse protein) | srIκBα | This paper | | Substituted serines 32 and 36 with alanines |
| Gene (*Mus musculus*) | Smad7 | UniProt | O35253-1 [1998-01-01 v1] | Amino acid sequence |
| Gene (*Mus musculus*) | SOCS1 | UniProt | O35716 [1998-01-01 v1] | |
| Software, algorithm | GenSmart | GenScript | RRID:SCR_026296 | Used to codon optimize amino acid sequences for mouse expression |
| Recombinant DNA reagent | VEE-mScarlet3-IRES-EGFP-IRES-moxBFP | This paper | RRID:Addgene_242407 | Deposited with Addgene: 242407 |
| Sequence-based reagent | Conventional saRNA construct | This paper | | saRNA transcribed from VEE-mScarlet3-IRES-EGFP-IRES-moxBFP |
| Recombinant DNA reagent | VEE-mScarlet3-IRES-EGFP-IRES-E3 | This paper | RRID:Addgene_242408 | Deposited with Addgene: 242408 |
| Sequence-based reagent | E3 saRNA construct | This paper | | saRNA transcribed from VEE-mScarlet3-IRES-EGFP-IRES-E3 |
| Recombinant DNA reagent | VEE-mScarlet3-IRES-EGFP-IRES-E3-NSs-L* | This paper | RRID:Addgene_242409 | Deposited with Addgene: 242409 |
| Sequence-based reagent | E3-NSs-L* saRNA construct | This paper | | saRNA transcribed from VEE-mScarlet3-IRES-EGFP-IRES-E3-NSs-L* |
| Recombinant DNA reagent | VEE-mScarlet3-IRES-E3-NSs-L*-EGFP-IRES-moxBFP | This paper | RRID:Addgene_242410 | Deposited with Addgene: 242410 |
| Sequence-based reagent | moxBFP saRNA construct | This paper | | saRNA transcribed from VEE-mScarlet3-IRES-E3-NSs-L*-EGFP-IRES-moxBFP |
| Recombinant DNA reagent | VEE-mScarlet3-IRES-E3-NSs-L*-EGFP-IRES-srIκBα | This paper | RRID:Addgene_242411 | Deposited with Addgene: 242411 |
| Sequence-based reagent | srIκBα saRNA construct | This paper | | saRNA transcribed from VEE-mScarlet3-IRES-E3-NSs-L*-EGFP-IRES-srIκBα |
| Recombinant DNA reagent | VEE-mScarlet3-IRES-E3-NSs-L*-EGFP-IRES-srIκBα-Smad7-SOCS1 | This paper | RRID:Addgene_242412 | Deposited with Addgene: 242412 |
| Sequence-based reagent | srIκBα-Smad7-SOCS1 saRNA construct | This paper | | saRNA transcribed from VEE-mScarlet3-IRES-E3-NSs-L*-EGFP-IRES-srIκBα-Smad7-SOCS1 |
| Recombinant DNA reagent | VEE-mScarlet3-IRES-PuroR | This paper | RRID:Addgene_242415 | Deposited with Addgene: 242415 |
| Recombinant DNA reagent | VEE-EGFP-IRES-PuroR | This paper | RRID:Addgene_242414 | Deposited with Addgene: 242414 |
| Recombinant DNA reagent | VEE-moxBFP-IRES-PuroR | This paper | RRID:Addgene_242413 | Deposited with Addgene: 242413 |
| Commercial assay, kit | Phusion High-Fidelity DNA Polymerase | Thermo Scientific | F530S | |

*Continued on next page*

*Continued*

| Reagent type (species) or resource | Designation | Source or reference | Identifiers | Additional information |
|---|---|---|---|---|
| Commercial assay, kit | NEBuilder HiFi DNA Assembly Cloning Kit | NEB | E5520S | |
| Strain, strain background (*Escherichia coli*) | 5-alpha Competent *E. coli* (High Efficiency) | NEB | C2987H | |
| Commercial assay, kit | T7 RiboMAX Large Scale RNA Production System | Promega | P1300 | |
| Commercial assay, kit | PureLink HiPure Plasmid FP Maxiprep kit | Invitrogen | K210027 | |
| Commercial assay, kit | Vaccinia Capping System | NEB | M2080S | |
| Commercial assay, kit | mRNA Cap 2′-O-Methyltransferase | NEB | M0366S | |
| Commercial assay, kit | Antarctic Phosphatase | NEB | M0289S | |
| Other | NorthernMax-Gly Sample Loading Dye | Invitrogen | AM8551 | |
| Other | Lipofectamine MessengerMAX Transfection Reagent | Invitrogen | LMRNA001 | |
| Strain, strain background (*M. musculus*) | C57BL/6J | Envigo | | |
| Cell line (*Homo sapiens*) | tSA201 | ECACC | 96121229 (RRID:CVCL_2737) | |
| Other | BioTracker NIR680 | Merck | SCT112 | |
| Chemical compound, drug | ML336 | Cayman Chemical | 9001920 | |
| Chemical compound, drug | Puromycin | Thermo Scientific | J67236.XF | |
| Other | 6-well black glass bottom plate | CellVis | P06-1.5H-N | |
| Other | 24-well black glass bottom plate | Grenier Bio-One | 662892 | |
| Other | Live Cell Imaging Solution | Invitrogen | A59688DJ | |
| Other | Poly-D-lysine–coated glass-bottom 35 mm dishes | MatTek | P35GC-1.5-14-C | |
| Software, algorithm | Linear unmixing script for Odyssey M images | This paper; **Ferguson, 2025** | | Available at https://github.com/lariferg/spectral_unmixing |
| Software, algorithm | ImageJ | NIH | RRID:SCR_003070 | |
| Other | Annexin V-CF800 | Biotium | 29078 | |
| Antibody | Anti-cadherin-11 rabbit polyclonal | Affinity Biosciences | DF3523 (RRID:AB_2835743) | |
| Antibody | Anti-Phospho-eIF2α (Ser51) rabbit monoclonal | Cell Signaling Technology | 3398 (RRID:AB_2096481) | |
| Antibody | Anti-eIF2α mouse monoclonal | Cell Signaling Technology | 2103 (RRID:AB_836874) | |
| Antibody | Anti-PKR rabbit polyclonal | Proteintech | 18244-1-AP (RRID:AB_2246451) | |
| Antibody | Anti-Phospho-eIF4E (S209) rabbit monoclonal | Abcam | ab76256 (RRID:AB_1523534) | |
| Antibody | Anti-eIF4E mouse monoclonal | Invitrogen | MA1-089 (RRID:AB_2536738) | |
| Antibody | Anti-fibroblast activation protein mouse monoclonal | InVivoMAb | BE0374 (RRID:AB_2927511) | |
| Antibody | Anti-puromycin mouse monoclonal | Absolute Antibody | 3RH11 (RRID:AB_2620162) | |
| Antibody | Goat anti-mouse IRDye 800CW secondary | LI-COR | 926-32210 (RRID:AB_621842) | |

*Continued on next page*

*Continued*

| Reagent type (species) or resource | Designation | Source or reference | Identifiers | Additional information |
|---|---|---|---|---|
| Antibody | Donkey anti-rabbit IRDye 800CW secondary | LI-COR | 926-32213 (RRID:AB_621848) | |
| Antibody | Donkey anti-mouse IRDye 680RD secondary | LI-COR | 926-68072 (RRID:AB_10953628) | |
| Other | CellTag 700 | LI-COR | 926-41090 | |
| Commercial assay, kit | DyLight Antibody Labeling Kit | Thermo Scientific | 53062 | |
| Software, algorithm | Empiria Studio | LI-COR | RRID:SCR_014281 | |
| Commercial assay, kit | LEGENDplex Mouse Anti-Virus Response Panel | BioLegend | 740621 | |
| Software, algorithm | Qognit | BioLegend | | |
| Other | 3D printed vacuum manifold for LEGENDplex filter plates | This paper | | Available at the NIH 3D Print Exchange under accession number 3DPX-021388 |
| Commercial assay, kit | RNeasy Mini Kit | QIAGEN | 74104 | |
| Commercial assay, kit | TaqMan Fast Advanced Cells-to-CT Kit | Invitrogen | A35377 | |
| Commercial assay, kit | TaqMan array plates | Invitrogen | 4413261 | |
| Software, algorithm | StepOne Software | Life Technologies | RRID:SCR_014281 | |
| Software, algorithm | Prism | GraphPad | RRID:SCR_002798 | |

## Plasmid cloning and construct design

The commercially available TagGFP2 E3L Simplicon vector (SCR725, Merck) served as the saRNA backbone in this study. After removal of the TagGFP2-IRES-E3 sequence, various constructs were generated by combining different elements using restriction digests (FastDigest, Thermo Scientific), overlap extension PCR (Phusion, Thermo Scientific), and HiFi assembly (NEBuilder, New England Biolabs).

Genetic elements were sourced as follows: mScarlet3 from pDx_mScarlet3 (a gift from Dorus Gadella [Addgene plasmid #189754]) (*Gadella et al., 2023*), IRES-EGFP from pIRES2-EGFP (Clontech), and IRES-E3 and IRES-PuroR (puromycin resistance) from the original TagGFP2 E3L Simplicon vector.

Custom gene synthesis (GeneArt, Thermo Scientific) was used to produce IRES-moxBFP, IRES-E3-T2A-NSs-P2A-L*-E2A-mEGFP, and IRES-srIκBα-P2A-Smad7-T2A-SOCS1 sequences. The IRES sequence was derived from pIRES2-EGFP (Clontech). Other sequences were obtained from public databases: moxBFP (FPbase ID: SSTDU) (*Costantini et al., 2015*), mEGFP (FPbase ID: QKFJN), Vaccinia virus E3 protein (UniProt: P21605-1), Toscana virus NSs protein (UniProt: P21699), Theiler's virus L* protein (UniProt: P0DJX4), mouse Smad7 (UniProt: O35253-1), and mouse SOCS1 (UniProt: O35716). srIκBα was engineered from mouse IκBα (UniProt: Q9Z1E3) by substituting serines 32 and 36 with alanines. 2A peptide sequences were preceded by GSG linkers: T2A (UniProt: Q9YK87 [139–156]); P2A (UniProt: A0A077CZU0 [977-995]); E2A (UniProt: K9MZ26 [991–1010]). All synthesized sequences were codon-optimized for mouse expression (GenSmart, GenScript).

Plasmids were cloned in DH5α competent *Escherichia coli* (High Efficiency, New England Biolabs) and purified by maxiprep (PureLink HiPure, Invitrogen). All plasmid sequences were verified using nanopore whole-plasmid sequencing (Plasmidsaurus).

## RNA synthesis

Plasmids were linearized with XbaI (FastDigest, Thermo Scientific) at 37°C for 3 hr. The linear plasmids were purified by phenol-chloroform extraction followed by sodium acetate-ethanol precipitation. Uncapped RNA was synthesized in vitro using the T7 RiboMAX Large Scale RNA Production System (Promega) at 37°C for 2 hr. After purification by ammonium acetate precipitation, the RNA was denatured by heating to 65°C for 5 min before rapidly cooling on ice. Cap-1 structures were then generated using the Vaccinia Capping System in conjunction with mRNA cap 2'-*O*-methyltransferase (both from New England Biolabs) for 45 min at 37°C. Following another round of ammonium acetate

precipitation, the RNA was treated with Antarctic phosphatase (New England Biolabs) for 30 min at 37°C. After a final ammonium acetate precipitation, RNA was resuspended in THE RNA Storage Solution (Invitrogen). RNA concentration was quantified using an N60 NanoPhotometer (Implen) and adjusted to a final concentration of 0.5 μg/μL. RNA was aliquoted and stored at –80°C until further use.

The integrity of the in vitro transcribed RNA was assessed by denaturing agarose gel electrophoresis (*Figure 4—figure supplement 3c*). Constructs designed to inhibit dsRNA-sensing pathways (conventional saRNA, E3, and E3-NSs-L*) produced a single band at the expected size. In contrast, constructs targeting inflammatory signaling (moxBFP, srIκBα, and srIκBα-Smad7-SOCS1) displayed two bands: one corresponding to the full-length transcript and a smaller, low-intensity band of consistent size across all three constructs (*Figure 4—figure supplement 3d*). The presence of this truncated species likely reflects premature termination at a cryptic terminator introduced during construct assembly.

## Denaturing RNA gel electrophoresis

In vitro transcribed RNA (1 μg) was mixed with 5 μL of NorthernMax-Gly Sample Loading Dye (Invitrogen) and 5 μL of nuclease-free water for glyoxal-based denaturation. The RNA ladder (RiboRuler High Range RNA Ladder, Thermo Scientific) was prepared by combining 2 μL of ladder, 3 μL of water, and 5 μL of loading dye. A loading dye-only control was also prepared using 5 μL of water and 5 μL of loading dye. All samples were incubated at 50°C for 30 min, then cooled on ice (*Floor, 2019*; *Rio, 2015*).

Samples were then loaded onto a 0.8% agarose gel prepared with diethyl pyrocarbonate-treated electrophoresis buffer consisting of 100 mM PIPES, 300 mM Bis-Tris, and 10 mM EDTA. Electrophoresis was performed at 5V/cm, and gels were imaged using an Odyssey Fc imager (LI-COR) with acquisition in the 600 nm channel over 10 min.

## Primary mouse FLS culture

Mouse tissue collection was conducted in accordance with Schedule 1 of the Animals (Scientific Procedures) Act 1986 Amendment Regulations 2012 and under Project Licence PP5814995 (granted to Ewan St. John Smith by the UK Home Office), with approval from the University of Cambridge Animal Welfare Ethical Review Body.

Primary FLS cultures were derived from wild-type C57BL/6J mice (Envigo, Bicester, UK), aged 5–12 weeks. Both male and female mice were used, although sex-based differences were not examined. Animals were group housed (≤5 per cage) in a temperature-controlled room (21°C) on a 12 hr light/dark cycle, with ad libitum access to food and water. Bedding, a red shelter, and enrichment materials were provided. Mice were euthanized via rising concentrations of $CO_2$ gas, and death was confirmed by dislocation of the atlanto-occipital joint using digital pressure to the lateral neck.

FLS were isolated from patellar explants used a modified version of a previously described protocol (*Chakrabarti et al., 2020*). Briefly, knee joints were exposed by incising the overlying skin with dissecting scissors. The patellar tendon was grasped near the patella using Dumont tweezers and severed with surrounding tissues using spring scissors. The patella was then excised by cutting the quadriceps tendon, and any excess muscle or tendons was carefully removed.

Both patellae from each animal were placed into a microcentrifuge tube containing ice-cold sterile phosphate-buffered saline (PBS), then transferred to a 24-well tissue culture plate (Costar) containing FLS culture medium: DMEM/F-12 (Invitrogen) supplemented with 25% fetal bovine serum (Sigma), 1×GlutaMAX (Gibco), and 100 μg/mL Primocin (InvivoGen). Explants were maintained in a humidified incubator at 37°C with 5% $CO_2$, with media changes every 1–2 days.

FLS outgrowth typically reached ~70% confluence by around 10 days in culture. At this point, the patellae were transferred to fresh wells for continued cell harvest. Adherent cells from the initial wells were dissociated using 0.1% trypsin-EDTA (Sigma) and passaged into a single well of a six-well plate (Costar). Cells were subsequently expanded into T25 flasks (BioLite, Thermo Scientific or CELLSTAR, Greiner Bio-One), with passaging upon confluence. From passage 2 onward, Versene solution (Gibco) was used for cell dissociation. All experiments used FLS between passages 3 and 8.

For downstream assays, cells from each animal were divided across multiple wells of 6-well or 24-well plates to permit within-animal, repeated-measures analysis. In cases where individual yields

were insufficient, cells from multiple animals were pooled prior to plating. Media was refreshed every 2–3 days following plating.

## tSA201 culture

tSA201 cells (96121229, ECACC) were cultured in media consisting of DMEM (Invitrogen) supplemented with 10% fetal bovine serum (Sigma), 1× nonessential amino acids (Gibco), 1 mM sodium pyruvate (Gibco), 1× GlutaMAX (Gibco), and 100 µg/mL normocin (InvivoGen). Adherent cultures were incubated in a humidified incubator at 37°C with 5% $CO_2$. Cultures were confirmed to be free of mycoplasma (MycoStrip, Invivogen).

## Immunocytochemistry and confocal microscopy

Cells were plated on poly-D-lysine-coated glass-bottom 35 mm dishes (P35GC-1.5-14-C, MatTek) and fixed with 4% paraformaldehyde for 10 min at room temperature without permeabilization. After fixation, cells were washed twice with PBS, followed by blocking with 10% normal goat serum in PBS for 1 hr at room temperature.

Cells were then incubated overnight at 4°C in blocking buffer, either without primary antibody (no primary control) or with cadherin-11 (extracellular) rabbit polyclonal antibody (1:200, DF3523, Affinity Biosciences). After three washes with PBS, cells were incubated for 1 hr at room temperature in blocking buffer containing 1 µg/mL Hoechst 33342, BioTracker NIR680 (1:2000, Merck), and goat anti-rabbit Alexa Fluor 488 secondary antibody (1:500, Invitrogen).

After four additional PBS washes, cells were imaged in PBS using a Leica Stellaris 5 confocal microscope equipped with a ×40 oil immersion objective. Tile-scanned images were stitched using Las X microscopy software (Leica) to generate high-resolution panoramic images.

## BioTracker staining

BioTracker NIR680 was diluted 1:2000 in unsupplemented DMEM/F-12. T25 flasks containing FLS were washed once with HBSS (with calcium and magnesium; Gibco), then incubated with the diluted dye for 30 min at 37°C. After incubation, the flasks were washed three times with FLS media, with each wash lasting 10 min at 37°C. FLS were then dissociated using Versene and plated onto 6- or 24-well plates.

## FLS transfection

For transfection in six-well plates, the medium was removed and replaced with 1 mL of Opti-MEM I (Gibco). In a microcentrifuge tube, 500 ng of saRNA was diluted in 200 µL of Opti-MEM I. In a separate tube, 3 µL of Lipofectamine MessengerMAX (Invitrogen) was diluted in 100 µL of Opti-MEM I. After gentle mixing, the solutions were combined and incubated at room temperature for 5 min, then added dropwise to the cells. The cells were incubated with the complexes for 2 hr at 37°C, after which the medium was removed and replaced with fresh FLS media. For transfection in 24-well plates, all volumes and amounts of saRNA were reduced fivefold.

## Microplate imaging

Black glass-bottom 6-well plates (P06-1.5H-N, Cellvis) or 24-well plates (662892, Greiner Bio-One) were used for microplate imaging. Prior to imaging, the FLS media was replaced with Live Cell Imaging Solution (Invitrogen). Imaging was performed using an Odyssey M laser scanner (LI-COR), using LI-COR acquisition software, a plate offset of +1.45 mm, and 100 µm resolution.

EGFP, mScarlet3, and BioTracker NIR680 were imaged using the 488, 520, and 700 channels, respectively. Spectral cross-excitation between EGFP and mScarlet3 in the 488 and 520 channels was corrected through linear unmixing analysis, as described below. Fluorescence intensity was quantified in ImageJ by measuring the integrated density within equal-area regions of interest for each well. Fluorescence values (EGFP, mScarlet3, and BioTracker) were normalized to the day 0 BioTracker signal measured before transfection to account for variations in starting cell number.

## Linear unmixing analysis

saRNA constructs encoding individual fluorescent proteins (moxBFP, EGFP, or mScarlet3) were designed to assess cross-excitation among the fluorescent proteins used in this study. tSA201 cells

were seeded in black glass-bottom six-well plates coated with poly-D-lysine (Sigma) and transfected with the respective saRNA constructs. Cells were imaged the following day using an Odyssey M laser scanner, with fluorescence captured through the 488 and 520 channels.

Expression of moxBFP was undetectable in both the 488 and 520 channels, and therefore it was excluded from further analysis. Bleed-through of EGFP into the 520 channel and mScarlet3 into the 488 channel was quantified using ImageJ, with cross-excitation determined to be 11.32% and 0.94%, respectively. The emission signals were assumed to be linearly proportional to the sum of the intensities of each fluorophore, and the unmixed EGFP and mScarlet3 signals were calculated using Python.

## Annexin V assay

FLS were stained with BioTracker NIR680 and plated on black glass-bottom 24-well plates (Sensoplate, Greiner Bio-One). Cells were transfected with saRNA or treated with 0.5 µM staurosporine (Cayman Chemical) as a positive control.

Annexin V-CF800 conjugate (Biotium) was diluted to 250 ng/mL in Live Cell Imaging Solution (Invitrogen). On each day of the assay, wells were washed with Live Cell Imaging Solution and replaced with diluted Annexin V-CF800 solution. Plates were incubated at 37°C for 15 min. Following incubation, plates were washed three times with Live Cell Imaging Solution before imaging the 700 and 800 channels with an Odyssey M imager set to +1.45 mm image offset and 100 µm resolution.

Image analysis was performed using ImageJ. To correct for unidirectional spectral bleed-through of BioTracker NIR680 into the 800-channel, subtractive compensation was applied by dividing the 700-channel image by a factor of 800 and then subtracting it from the 800-channel image. Due to the presence of a small number of high-intensity speckles in the 800-channel image, area rather than fluorescence intensity was quantified. The display range of the 800-channel image was set between 0.25 and 2.5, and the image was thresholded using the method of Li (*Li and Lee, 1993*). The area of thresholded pixels within each well was then quantified and adjusted based on the BioTracker signal before transfection to account for variations in cell number. Data were then normalized to the average of the mock transfection condition.

## Calcein AM staining

Calcein AM (Invitrogen) staining was performed according to the manufacturer's protocol. Briefly, cells were washed with HBSS and incubated with 2 µM calcein AM (diluted in HBSS) at 37°C for 30 min. Following three washes, cells were imaged in Live Cell Imaging Solution, and the 488-channel image was captured using an Odyssey M imager.

Although calcein AM and EGFP have overlapping spectra, calcein AM fluorescence was typically much higher than EGFP, making EGFP's contribution to the measured Calcein AM signal negligible in most cases. However, when the E3 construct was used, EGFP expression was sufficient to interfere with the signal. To address this, in experiments that included use of the E3 construct, a 488-channel image was captured prior to calcein AM application and subtracted from the post-application image to accurately measure calcein AM fluorescence.

Fluorescence intensity was quantified using ImageJ. In experiments where BioTracker was used, fluorescence values were corrected by the pre-transfection BioTracker signal to account for differences in starting cell number.

## In-cell western assay

Cells were plated on black, glass-bottom 24-well plates. Alongside mock and saRNA transfections, one well was reserved for background subtraction, which received no treatment. Unless specified otherwise, experiments were conducted on day 2 post-transfection. Cells were fixed with 4% paraformaldehyde for 10 min at room temperature, followed by two washes with Tris-buffered saline (TBS). Permeabilization was then performed using 0.1% Triton X-100 in TBS for 10 min, followed by two additional TBS washes. After permeabilization, cells were blocked with Intercept TBS Blocking Buffer (LI-COR) for 1 hr at room temperature with gentle agitation.

Primary antibodies were diluted in Intercept Blocking Buffer and incubated with the cells overnight at 4°C. The background subtraction well was incubated with blocking buffer alone, without primary antibodies. The following primary antibodies were used: Phospho-eIF2α (Ser51) (D9G8) XP rabbit monoclonal (1:200, #3398, Cell Signaling Technology), eIF2α (L57A5) mouse monoclonal

(1:200, #2103, Cell Signaling Technology), PKR rabbit polyclonal (1:500, #18244-1-AP, Proteintech), Phospho-eIF4E (S209) rabbit monoclonal (1:200, #ab76256, Abcam), eIF4E mouse monoclonal (5D11) (1:200, #MA1-089, Invitrogen), and FAP (73.3) mouse monoclonal (20 µg/mL, #BE0374, InVivoMAb). All primary antibodies were unconjugated, except for the FAP antibody, which was conjugated to DyLight 800 using the DyLight Antibody Labeling Kit (#53062, Thermo Scientific) according to manufacturer's protocols.

After washing the wells four times with TBS supplemented with 0.1% Tween 20 (TBST), cells were incubated for 1 hr at room temperature with the appropriate secondary antibodies or normalization stain, with gentle agitation. The secondary antibodies used were goat anti-mouse IRDye 800CW (1:800), donkey anti-rabbit IRDye 800CW (1:800), donkey anti-mouse IRDye 680RD (1:800), and CellTag 700 (1:500), all from LI-COR. The background subtraction well received secondary antibody but no CellTag. Following four additional washes with TBST, the plate was inverted and gently tapped on absorbent paper to remove excess liquid. The plate was then imaged using the Odyssey M imager with LI-COR Acquisition software using a plate offset of +1.45 and 100 µm resolution. Signal quantification was carried out using Empiria Studio software (LI-COR).

## SUnSET assay

FLS were plated in black, glass-bottom 24-well plates, with one well reserved for background subtraction (no treatment). Two days post-transfection, cells were treated with 1 µM puromycin (Sigma) for 30 min at 37 °C. The background control well did not receive puromycin. Following incubation, cells were washed twice with HBSS and fixed in 5% neutral-buffered formalin (in PBS) for 20 min at room temperature. Cells were then washed three times with PBS, permeabilized with 0.2% Triton X-100 for 10 min, and blocked with Intercept Blocking Buffer for 1 hr at room temperature.

Anti-puromycin antibody (3RH11, Absolute Antibody) was diluted 1:250 in blocking buffer and incubated with cells overnight at 4 °C. The following day, cells were washed four times with TBST, then incubated with goat anti-mouse IRDye 800CW (1:800) and CellTag 700 (1:500) diluted in blocking buffer. The background subtraction well did not receive CellTag. After four additional washes with TBST, plates were inverted, tapped dry on absorbent paper, and imaged using the Odyssey M imager. Signal quantification was performed using Empiria Studio software.

## rRNA integrity analysis

FLS were plated in six-well plates and either transfected with saRNA or subjected to mock transfection. After 26–30 hr (allowing for saRNA replication but before substantial cell loss), cells were harvested by scraping in 350 µL RLT buffer (QIAGEN) supplemented with 10 µL/mL β-mercaptoethanol (Sigma). Cell lysates were homogenized using QIAshredder columns (QIAGEN), and total RNA was extracted using the RNeasy Mini Kit (QIAGEN) according to the manufacturer's instructions. RNA quantity and quality were initially assessed using a N60 nanophotometer (Implen). Samples were aliquoted and stored at –80°C until further analysis. For detailed quality assessment, samples were transported on dry ice and analyzed using the RNA Pico Kit on the 2100 Bioanalyzer System (Agilent), performed by Cambridge Genomic Services. RIN values were determined for each sample.

## Multiplex bead-based immunoassay

FLS were stained with BioTracker and seeded onto six-well plates. After transfection, the saRNA-lipofectamine complexes were replaced with 2 mL of fresh FLS media. Two days later, supernatants were collected and stored at –80°C until analysis. On the day of analysis, the supernatant was thawed on ice, and levels of 13 cytokines (IFN-γ, CXCL1, TNF-α, MCP-1, IL-12p70, CCL5, IL-1β, CXCL10, GM-CSF, IL-10, IFN-β, IFN-α, and IL-6) were measured using the LEGENDplex Mouse Anti-Virus Response Panel (740621, BioLegend) following the manufacturer's protocol. Samples were run in duplicate. A custom 3D-printed PETG vacuum manifold (designed with Fusion 360) was used for vacuum filtration, and beads were analyzed using a CytoFLEX LX flow cytometer (Beckman Coulter). Data analysis was performed with the LEGENDplex Qognit Data Analysis Suite (BioLegend). To account for differences in cell number between wells, cytokine levels were normalized to the BioTracker signal before transfection.

## ML336 experiments

ML336 (Cayman Chemical) was dissolved in DMSO (Sigma) to a stock concentration of 10 mM and stored in aliquots at –20°C. ML336 was added to cell cultures at the indicated final concentrations. Vehicle controls received either 0.01% or 0.1% DMSO, corresponding to the maximum DMSO concentrations used in the respective ML336-treated conditions.

In concentration–response experiments, ML336 was applied during transfection. mScarlet3 fluorescence was measured on day 2 post-transfection. Data were fit to a variable-slope sigmoidal model, and a single data point was excluded due to pipetting error. In washout experiments, media was replaced 3 times on day 2 post-transfection and mScarlet3 signal was measured the following day. In these experiments, data were fit to a bell-shaped concentration–response curve.

## RT-qPCR experiments

FLS were seeded in 96-well plates, and transfections were performed using a 30-fold downscaling of the volumes used in six-well formats. At 2 days post-transfection, cDNA was synthesized directly from cell lysates using the TaqMan Fast Advanced Cells-to-CT Kit (Invitrogen), according to the manufacturer's protocol and stored at –80 °C until analysis. For each biological replicate, a single well that received no treatment and was processed without reverse transcriptase (RT) served as a shared no-RT control.

For each treatment condition, a reaction mix containing cDNA, RT-PCR grade water (Invitrogen), and TaqMan Fast Advanced Master Mix (Applied Biosystems) was prepared and dispensed into the corresponding wells of custom 96-well 0.1 mL TaqMan array plates (Thermo Fisher). All conditions for a given biological replicate were run on the same plate to minimize inter-plate variability. Plates were run in duplicate, and each plate included no-template and no-RT controls, along with cDNA from mock-transfected cells, and cells transfected with conventional saRNA, E3, or srIκBα-Smad7-SOCS1 constructs.

Each array included the following TaqMan Gene Expression Assays: *18S* (Hs99999901_s1), *Adar* (Mm00508001_m1), *Isg20* (Mm00469585_m1), *Rigi* (Mm01216853_m1), *Ifih1* (Mm00459183_m1), *Oas1e* (Mm00653062_m1), *Tlr3* (Mm01207404_m1), *Eif2ak2* (Mm01235643_m1), *ZC3hav1* (Mm00512227_m1), *Rnasel* (Mm00712008_m1), *EGFP* (Mr04329676_mr), *Ifnb1* (Mm00439552_s1), *Ifna2* (Mm00833961_s1), *Ifng* (Mm01168134_m1), *Tnf* (Mm00443258_m1), and *Il6* (Mm00446190_m1).

Reactions were run for 40 cycles on a StepOnePlus Real-Time PCR System (Applied Biosystems), using manufacturer-recommended conditions. A consistent ΔRn threshold was applied across all plates and replicates. ΔCT values were calculated relative to *18S*. Transcripts not detected in all samples were excluded from analysis, with the exception of EGFP. These excluded transcripts included *Oas1e*, *Ifnb1*, *Ifna2*, *Ifng*, and *Tnf*. Statistical significance was evaluated using ΔCT values, and mean –ΔΔCT values relative to the mock condition are shown as a heatmap for visualization.

## Statistical analysis

All statistical analyses and curve fitting were conducted using GraphPad Prism 9, with specific tests detailed in the corresponding figure legends. All statistical tests were two-sided tests.

## Materials availability statement

All key plasmids generated in this study have been deposited with the Addgene repository. Accession numbers have been provided in the Key Resources Table.

# Acknowledgements

The authors gratefully acknowledge the Cambridge Advanced Imaging Centre, the flow cytometry facility from the School of the Biological Sciences, and Cambridge Genomic Services for their support &and assistance in this work. The authors would like to thank Dr. Paul Miller for providing the pDx_mScarlet3 plasmid and Dr. Alex Cloake for providing the pIRES2-EGFP plasmid used in this study. TKYL acknowledges support from a Horizon Europe Marie Skłodowska-Curie Actions European Postdoctoral Fellowship (UKRI Guarantee) (EP/X023117/1). AR and LWP disclose support from Astra-Zeneca PhD studentships (G115018 and G113502, respectively). LF discloses support from funding provided by the MRC (MC-A023-5PB91). EStJS acknowledges funding from the UKRI and Versus

Arthritis (MR/W002426/1) as part of the ADVANTAGE visceral pain consortium through the Advanced Pain Discovery Platform (APDP) and EStJS and LJG acknowledge the Wellcome Trust (225856/Z/22/Z). The eLife publication fees for this article were supported by the University of Cambridge's open access funding.

## Additional information

### Funding

| Funder | Grant reference number | Author |
|---|---|---|
| Marie Skłodowska-Curie Actions | European Postdoctoral Fellowship (UKRI Guarantee EP/X023117/1) | Tony KY Lim |
| AstraZeneca PLC | PhD studentship G115018 | Anne Ritoux |
| AstraZeneca PLC | PhD studentship G113502 | Luke W Paine |
| Medical Research Council | MC-A023-5PB91 | Larissa Ferguson |
| UK Research and Innovation | MICA ADVANTAGE visceral pain consortium MR/W002426/1 | Ewan St John Smith |
| Versus Arthritis | MICA ADVANTAGE visceral pain consortium MR/W002426/1 | Ewan St John Smith |
| Wellcome | 10.35802/225856 | Laura J Grundy Ewan St John Smith |

The funders had no role in study design, data collection and interpretation, or the decision to submit the work for publication. For the purpose of Open Access, the authors have applied a CC BY public copyright license to any Author Accepted Manuscript version arising from this submission.

### Author contributions

Tony KY Lim, Conceptualization, Resources, Formal analysis, Supervision, Funding acquisition, Investigation, Visualization, Methodology, Writing – original draft, Writing – review and editing; Anne Ritoux, Luke W Paine, Laura J Grundy, Investigation; Larissa Ferguson, Software, Writing – review and editing; Tawab Abdul, Methodology; Ewan St John Smith, Conceptualization, Resources, Formal analysis, Supervision, Funding acquisition, Visualization, Writing – review and editing

### Author ORCIDs

Tony KY Lim ⓘD https://orcid.org/0000-0003-1843-0060
Luke W Paine ⓘD https://orcid.org/0009-0009-4099-4648
Larissa Ferguson ⓘD https://orcid.org/0000-0003-4274-8634
Laura J Grundy ⓘD https://orcid.org/0009-0003-8938-0862
Ewan St John Smith ⓘD https://orcid.org/0000-0002-2699-1979

### Ethics

Mouse tissue collection was conducted in accordance with Schedule 1 of the Animals (Scientific Procedures) Act 1986 Amendment Regulations 2012 and under Project Licence PP5814995 (granted to Ewan St. John Smith by the UK Home Office), with approval from the University of Cambridge Animal Welfare Ethical Review Body.

Reviewer #1 (Public review): https://doi.org/10.7554/eLife.105978.3.sa1
Reviewer #3 (Public review): https://doi.org/10.7554/eLife.105978.3.sa2
Author response https://doi.org/10.7554/eLife.105978.3.sa3

## Additional files

**Supplementary files**
MDAR checklist

**Data availability**
All data generated in this study have been deposited in the Figshare repository and are publicly available at http://doi.org/10.6084/m9.figshare.27091972. Whole-plasmid sequencing results of the key plasmids generated in this study are provided in the article's source data files. The Python code used for the spectral unmixing analysis is available on GitHub (https://github.com/lariferg/spectral_unmixing, copy archived at *Ferguson, 2025*). The 3D model for the 3D printed vacuum manifold designed for LEGENDplex filter plates is available from the NIH 3D Print Exchange under accession number 3DPX-021388. All other relevant data are included within the article, appendix, and associated source data files.

The following datasets were generated:

| Author(s) | Year | Dataset title | Dataset URL | Database and Identifier |
|---|---|---|---|---|
| Lim TKY | 2025 | Immune-Evasive Self-amplifying RNA Complete Dataset (eLife 2025) | http://doi.org/10.6084/m9.figshare.27091972 | figshare, 10.6084/m9.figshare.27091972 |
| Lim TKY | 2024 | Vacuum manifold for BioLegend LEGENDplex bead-based immunoassay (96-well filter plate) | https://3d.nih.gov/entries/3DPX-021388 | NIH 3D, 3DPX-021388 |

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

## Appendix 1

### Development and validation of a microplate assay for longitudinal monitoring of cap-dependent and cap-independent transgene expression and cell number

Self-amplifying RNA (saRNA) replication can trigger innate immune responses, leading to shutdown of cap-dependent translation and cytotoxicity (*Frolov and Schlesinger, 1994*; *Venticinque and Meruelo, 2010*; *Mastrangelo et al., 2000*; *Frolov et al., 1999*), affecting transgene expression. To monitor subgenomic-driven (cap-dependent) and IRES-driven (cap-independent) transgene expression and cell number longitudinally within live cells over several weeks, we developed a novel microplate-based assay. This approach enables higher throughput than microscopy-based assays, avoids destructive, single time-point methods, and eliminates the labor-intensive sample preparation required for viability staining or western blotting.

Our assay utilizes dual-reporter saRNA constructs (main text, *Figure 1a*) that co-express at least two fluorescent proteins. These constructs report on two distinct modes of translation: mScarlet3 is expressed from the subgenomic RNA via cap-dependent initiation, while EGFP is expressed from an EMCV IRES to report on cap-independent initiation. To concurrently track cell number, cells were pre-stained with BioTracker NIR680, a near-infrared lipophilic carbocyanine membrane dye for long-term fluorescent labeling of cells (*Honig and Hume, 1986*). This design allows for three parameters to be quantified in tandem using an Odyssey M microplate imager: cap-dependent transgene expression (mScarlet3, 520 channel), cap-independent transgene expression (EGFP, 488 channel), and cell number (BioTracker NIR680, 700 channel).

We validated the assay in primary mouse fibroblast-like synoviocytes (FLS) derived from patellar explants. The identity of these cells was first confirmed by positive immunofluorescent staining for the FLS marker Cadherin-11 (*Appendix 1—figure 1*; *Lee et al., 2007*; *Valencia et al., 2004*).

Next, we validated the use of BioTracker NIR680 for indicating cell number and compared its performance to the viability dye calcein AM (*Bratosin et al., 2005*). We observed that mock transfection with the lipid-based agent Lipofectamine MessengerMAX increased the baseline fluorescence of BioTracker NIR680 compared to non-transfected cells (*Appendix 1—figure 2a and b*). This is likely due to the dye interacting with the lipid components of the transfection reagent, as carbocyanine dyes fluoresce brightly upon integration into lipid bilayers.

To confirm that BioTracker NIR680 fluorescence still reliably reported changes in cell number despite this baseline shift, we treated mock-transfected cells with the apoptosis inducer staurosporine (*Bertrand et al., 1994*). This treatment caused a significant reduction in BioTracker NIR680 fluorescence, demonstrating its sensitivity to cell death (*Appendix 1—figure 2c and d*). In comparison, calcein AM fluorescence was unaffected by mock transfection but was similarly reduced by staurosporine. These results support the use of BioTracker NIR680 for tracking relative cell number, as it is spectrally distinct from our reporters and does not require post-transfection staining or wash steps.

Finally, we addressed the potential for spectral overlap between EGFP and mScarlet3 signals when measured with the Odyssey M imager. Using saRNA constructs expressing only EGFP or mScarlet3 (*Appendix 1—figure 3a*), we quantified the signal bleed-through between the 488 and 520 channels (*Appendix 1—figure 3b*). Based on this, we established and applied a linear unmixing algorithm to separate the two fluorescent signals in all subsequent experiments (*Appendix 1—figure 3c*). We also confirmed that a third fluorescent protein, moxBFP, used as a control in some experiments, produced no detectable signal in either the EGFP or mScarlet3 channels, ensuring no interference.

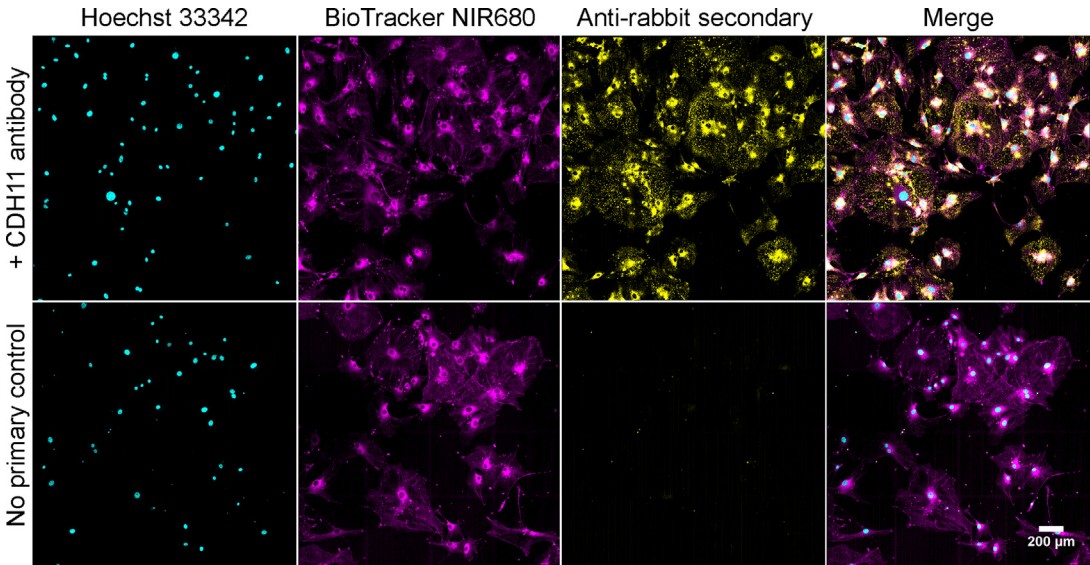

**Appendix 1—figure 1.** Confocal microscopy images confirming fibroblast-like synoviocyte (FLS) identity of primary cells isolated from mouse patellar explants. Cells were labelled with BioTracker NIR680 (lipophilic membrane dye) and Hoechst 33342 (nuclear stain). The top row shows cells treated with rabbit anti-cadherin-11 (CDH11) primary antibody, while the bottom row presents the no-primary control to verify specificity. Isolated cells exhibit positive CDH11 immunostaining, an established FLS marker. Scale bar = 200 μm.

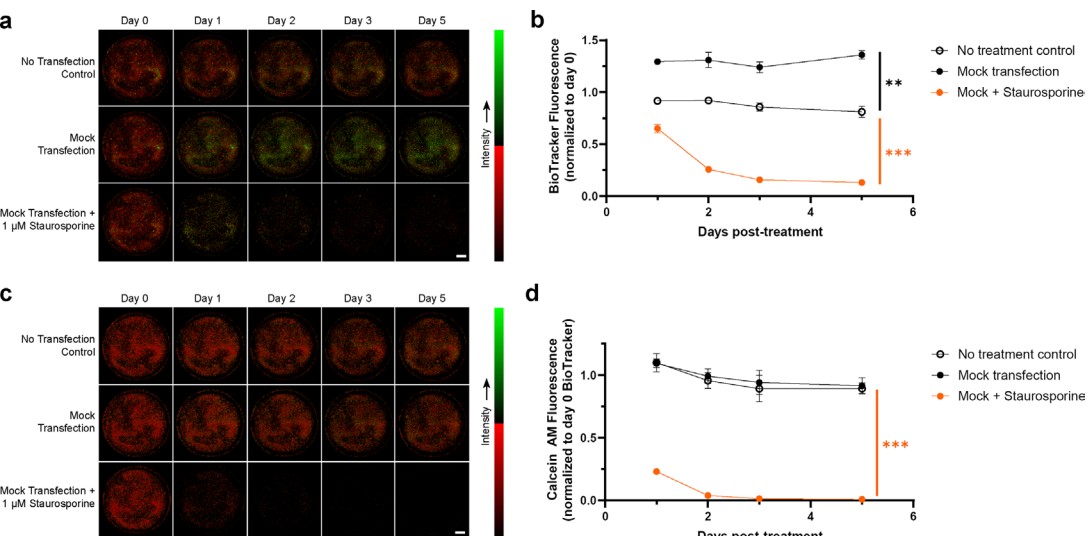

**Appendix 1—figure 2.** BioTracker fluorescence indicates cell number following staurosporine-induced apoptosis, despite an increase after mock transfection. (**a**) Representative microplate images of FLS stained with BioTracker under three conditions: no transfection control, mock transfection, and mock transfection with 1 μM staurosporine. Scale bar = 5 mm. (**b**) Quantification of BioTracker fluorescence (n=3). BioTracker signal increases after mock transfection but decreases following staurosporine treatment. (**c**) Representative images of the same FLS in panel (a) stained with calcein AM, a viability dye. Scale bar = 5 mm. (**d**) Quantification of calcein AM fluorescence (n=3). Calcein AM fluorescence does not increase after mock transfection but decreases after staurosporine treatment. For panels (b,d): Statistical significance was determined by two-way repeated-measures ANOVA with Dunnett's multiple comparisons test comparing groups to the no treatment control. **$p<0.01$ and ***$p<0.001$. Data are presented as mean ± standard error of the mean.

The online version of this article includes the following source data for appendix 1—figure 2:

**Appendix 1—figure 2—source data 1.** Numerical data used to generate the plots in *Appendix 1—figure 2*.

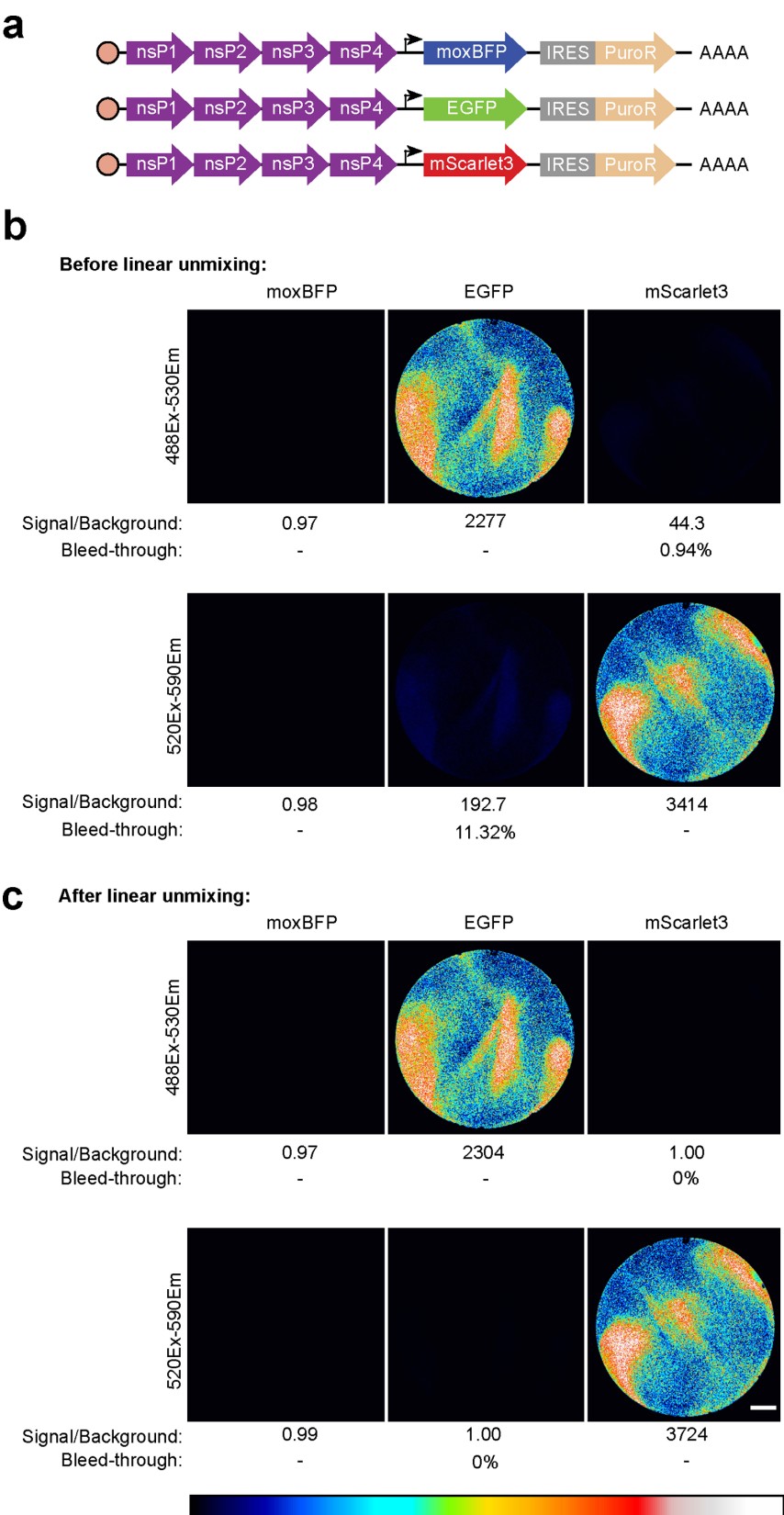

**Appendix 1—figure 3.** Linear unmixing corrects spectral overlap of EGFP and mScarlet3 fluorescent signals imaged using the Odyssey M laser scanner. (**a**) Schematic of the self-amplifying RNA constructs used for linear

*Appendix 1—figure 3 continued on next page*

*Appendix 1—figure 3 continued*

unmixing, each expressing a different fluorescent protein: moxBFP, EGFP, and mScarlet3. Constructs were transfected into tSA201 cells. (**b**) Representative microplate images showing fluorescence in the 488 and 520 channels, 1 day after transfection. moxBFP-transfected cells were not detected in either channel. EGFP was primarily detected in the 488 channel, with 11.32% bleed-through into the 520 channel. mScarlet3 was primarily detected in the 520 channel, with 0.94% bleed-through into the 488 channel. Background signal was determined from mock-transfected cells. (**c**) Corrected images using linear unmixing to counteract spectral bleed-through from mScarlet3 and EGFP signals. Scale bars = 5 mm.

The online version of this article includes the following source data for appendix 1—figure 3:

**Appendix 1—figure 3—source data 1.** Whole-plasmid sequencing results for self-amplifying RNA (saRNA) constructs depicted in *Appendix 1—figure 3a*.

