## [Editor Report · eLife Assessment]

In this manuscript, Lim and collaborators present an **important** system for developing self-amplifying RNA with **convincing** evidence that it does not provoke a strong host inflammatory response in cultured cells. This approach could be further strengthened going forward by testing these self-amplying RNAs in an in vivo system.

---

## [Referee Report · Reviewer #1 (Public review)]

Summary:

The authors have developed self-amplifying RNAs (saRNAs) encoding additional genes to suppress dsRNA-related inflammatory responses and cytokine release. Their results demonstrate that saRNA constructs encoding anti-inflammatory genes effectively reduce cytotoxicity and cytokine production, enhancing the potential of saRNAs. This work is significant for advancing saRNA therapeutics by mitigating unintended immune activation.

Strengths:

This study successfully demonstrates the concept of enhancing saRNA applications by encoding immune-suppressive genes. A key challenge for saRNA-based therapeutics, particularly for non-vaccine applications, is the innate immune response triggered by dsRNA recognition. By leveraging viral protein properties to suppress immunity, the authors provide a novel strategy to overcome this limitation. The study presents a well-designed approach with potential implications for improving saRNA stability and minimizing inflammatory side effects.

Comments on revisions:

All comments have been thoroughly addressed, and the manuscript has been significantly improved.

---

## [Referee Report · Reviewer #3 (Public review)]

Summary:

Context - this is the 2nd review, of a manuscript that has already undergone some revisions.

The manuscript explores ways to make self-amplifying RNA (saRNA) more silent through the inclusion of genes to inhibit the innate immune response. The readouts are predominantly expression and cell viability. They take a layered approach, adding multiple genes, as well as altering the capping of the anti-immune genes.

Strengths:

As described by the other reviewers, the authors take a stepwise approach to demonstrate that they can lead to sustained expression of the transgene.

Weaknesses:

The following weaknesses need some consideration

(1) The data show sustained expression, but do not directly show amplification. The amount of RFP is constantly decreasing over the time course. There is some evidence for the srIκBα-Smad7-SOCS1 construct. But measuring the RNA itself would be beneficial

(2) The end construct is very large - it has 12 genes, this may have manufacturing considerations, affecting the translatability.

---

## [Author Response]

The following is the authors’ response to the original reviews

**Public Reviews:**

**Reviewer #1 (Public review):**
Summary:The authors have developed self-amplifying RNAs (saRNAs) encoding additional genes to suppress dsRNA-related inflammatory responses and cytokine release. Their results demonstrate that saRNA constructs encoding anti-inflammatory genes effectively reduce cytotoxicity and cytokine production, enhancing the potential of saRNAs. This work is significant for advancing saRNA therapeutics by mitigating unintended immune activation.Strengths:This study successfully demonstrates the concept of enhancing saRNA applications by encoding immune-suppressive genes. A key challenge for saRNA-based therapeutics, particularly for non-vaccine applications, is the innate immune response triggered by dsRNA recognition. By leveraging viral protein properties to suppress immunity, the authors provide a novel strategy to overcome this limitation. The study presents a well-designed approach with potential implications for improving saRNA stability and minimizing inflammatory side effects.

We thank Reviewer #1 for their thorough review and for recognizing both the significance of our work and the potential of our strategy to expand saRNA applications beyond vaccines.

Weaknesses:(1) Impact on Cellular Translation:The authors demonstrate that modified saRNAs with additional components enhance transgene expression by inhibiting dsRNA-sensing pathways. However, it is unclear whether these modifications influence global cellular translation beyond the expression of GFP and mScarlet-3 (which are encoded by the saRNA itself). Conducting a polysome profiling analysis or a puromycin labeling assay would clarify whether the modified saRNAs alter overall translation efficiency. This additional data would strengthen the conclusions regarding the specificity of dsRNA-sensing inhibition.

We thank the Reviewer for this insightful suggestion. We performed a puromycin labeling assay to assess global translation rates (Figure 3—figure supplement 1c). This experiment revealed that the E3 construct significantly reduces global protein synthesis, despite driving high levels of saRNAencoded transgene expression (Figure 1d, e). In contrast, the E3-NSs-L* construct mitigated this reduction in global translation while maintaining moderate transgene expression. These findings support our hypothesis that E3 enhances transgene output in part by activating RNase L, which degrades host mRNAs and thereby reduces ribosomal competition. We appreciate the Reviewer’s recommendation of this experiment, which has strengthened the manuscript.

(2) Stability and Replication Efficiency of Long saRNA Constructs:The saRNA constructs used in this study exceed 16 kb, making them more fragile and challenging to handle. Assessing their mRNA integrity and quality would be crucial to ensure their robustness.Furthermore, the replicative capacity of the designed saRNAs should be confirmed. Since Figure 4 shows lower inflammatory cytokine production when encoding srIkBα and srIkBαSmad7-SOCS1, it is important to determine whether this effect is due to reduced immune activation or impaired replication. Providing data on replication efficiency and expression levels of the encoded anti-inflammatory proteins would help rule out the possibility that reduced cytokine production is a consequence of lower replication.

We thank the Reviewer for these valuable suggestions.

To assess the integrity of the saRNA constructs, we performed denaturing gel electrophoresis (Supplemental Figure 6c). The native saRNA, E3, and E3-NSs-L* constructs each migrated as a single band. The moxBFP, srIκBα, and srIκBα-Smad7-SOCS1 constructs showed both a full-length transcript and a lower-abundance truncated band (Supplemental Figure 6d), suggestive of a cryptic terminator sequence introduced in a region common to these three constructs.

To evaluate replicative capacity, we performed qPCR targeting EGFP, which is encoded by all constructs. This analysis revealed that the srIκBα-Smad7-SOCS1 construct exhibited lower replication efficiency than both native saRNA and E3. Several factors may contribute to this difference, including the longer transcript length, reduced molar input when equal mass was used for transfection, prevention of host mRNA degradation due to RNase L inhibition, or the presence of truncated transcripts.

Given these confounding variables, we revised our approach to analyzing cytokine production. Rather than comparing all six constructs together, we split the analysis into two parts: (1) the effects of dsRNA-sensing pathway inhibition (Figure 4a), and (2) the effects of inflammatory signalling inhibition (Figure 4c). For the latter, we compared srIκBα and srIκBα-Smad7-SOCS1 to moxBFP, as these three constructs are more comparable in size, share the same truncated transcript, and all encode L* to inhibit RNase L. This strategy minimizes the likelihood that differences in the cytokine responses are due to variation in replication efficiency.

(3) Comparative Data with Native saRNA:Including native saRNA controls in Figures 5-7 would allow for a clearer assessment of the impact of additional genes on cytokine production. This comparison would help distinguish the effect of the encoded suppressor proteins from other potential factors.

We thank the Reviewer for this helpful suggestion. We have added the native saRNA condition to Figure 5 as a visual reference. However, due to the presence of truncated transcripts in the constructs designed to inhibit inflammatory signalling pathways, the actual amount of full-length saRNA delivered in these conditions is likely lower than expected, despite using equal total RNA mass for transfection. This complicates direct comparisons with constructs targeting dsRNAsensing pathways, which do not show transcript truncation. For this reason, native saRNA was included only as a visual reference and was not used in statistical comparisons with the inflammatory signalling inhibitor constructs.

(4) In vivo Validation and Safety Considerations:Have the authors considered evaluating the in vivo potential of these saRNA constructs? Conducting animal studies would provide stronger evidence for their therapeutic applicability. If in vivo experiments have not been performed, discussing potential challenges - such as saRNA persistence, biodistribution, and possible secondary effectswould be valuable.(5) Immune Response to Viral Proteins:Since the inhibitors of dsRNA-sensing proteins (E3, NSs, and L*) are viral proteins, they would be expected to induce an immune response. Analyzing these effects in vivo would add insight into the applicability of this approach.

We appreciate the Reviewer’s points regarding in vivo validation and safety considerations. While in vivo studies are beyond the scope of the present investigation, we agree that evaluating therapeutic potential, biodistribution, persistence, and secondary effects will be essential for future translation. We have now included a brief discussion of these considerations at the end of the revised discussion. In ongoing work, we are planning follow-up studies incorporating in vivo imaging and functional assessments of saRNA-driven cargo delivery in preclinical models of inflammatory joint pain.

Regarding the immune response to viral proteins, we agree that this is an important consideration and have now included a clearer discussion of this limitation in the revised manuscript. Specifically, we highlight that encoding multiple viral inhibitors (E3, NSs, and L*), in combination with the VEEV replicase, may increase the likelihood of adaptive immune recognition via MHC class I presentation. This could lead to cytotoxic T cell–mediated clearance of saRNA-transfected cells, thereby limiting therapeutic durability. We emphasize that addressing both intrinsic cytotoxicity and immune-mediated clearance will be essential for advancing the clinical potential of this platform.

(6) Streamlining the Discussion Section:The discussion is quite lengthy. To improve readability, some content - such as the rationale for gene selection-could be moved to the Results section. Additionally, the descriptions of Figure 3 should be consolidated into a single section under a broader heading for improved coherence.

Thank you for these helpful suggestions. We have streamlined the Discussion to improve readability and have moved the rationale for gene selection to the results section, as recommended. In addition, we have consolidated the Figure 3 descriptions to improve coherence and to simplify the presentation.

**Reviewer #2 (Public review):**
Summary:Lim et al. have developed a self-amplifying RNA (saRNA) design that incorporates immunomodulatory viral proteins, and show that the novel design results in enhanced protein expression in vitro in mouse primary fibroblast-like synoviocytes. They test constructs including saRNA with the vaccinia virus E3 protein and another with E3, Toscana virus NS protein and Theiler's virus L protein (E3 + NS + L), and another with srIκBα-Smad7SOCS1. They have also tested whether ML336, an antiviral, enables control of transgene expression.Strengths:The experiments are generally well-designed and offer mechanistic insight into the RNAsensing pathways that confer enhanced saRNA expression. The experiments are carried out over a long timescale, which shows the enhance effect of the saRNA E3 design compared to the control. Furthermore, the inhibitors are shown to maintain the cell number, and reduce basal activation factor-⍺ levels.

We thank Reviewer #2 for their thoughtful and detailed assessment of our manuscript, and for recognizing the mechanistic insights provided by our study. We also appreciate their positive comments on the experimental design, the extended timescale, and the observed effects on transgene expression, cell viability, and basal fibroblast activation factor-α levels.

Weaknesses:One limitation of this manuscript is that the RNA is not well characterized; some of the constructs are quite long and the RNA integrity has not been analyzed. Furthermore, for constructs with multiple proteins, it's imperative to confirm the expression of each protein to confirm that any therapeutic effect is from the effector protein (e.g. E3, NS, L). The ML336 was only tested at one concentration; it is standard in the field to do a dose-response curve. These experiments were all done in vitro in mouse cells, thus limiting the conclusion we can make about mechanisms in a human system.

Thank you for your detailed feedback. We have added new experiments and clarified limitations in the revised manuscript to address these concerns:

RNA integrity: We performed denaturing gel electrophoresis on the in vitro transcribed saRNA constructs (Supplemental Figure 7c). Constructs targeting dsRNA-sensing pathways migrated as a single band, while those targeting inflammatory signalling pathways showed both a full-length product and a common, lower-abundance truncated transcript. This suggests that the actual amount of full-length RNA delivered for the constructs inhibiting inflammatory signalling was overestimated. To account for this, we avoided direct comparisons between the two types of constructs and instead focused on comparisons within each type to ensure more meaningful interpretation.

Confirmation of protein expression: While we acknowledge that direct measurement of each protein would provide additional insight, we believe the functional assays presented offer strong evidence that the encoded proteins are expressed and exert their intended biological effects. Additionally, IRES functionality was confirmed visually using fluorescent protein reporters, supporting the successful expression of downstream genes.

ML336 concentration–response: We have now performed a concentration–response analysis for ML336 (Figure 8a and b), which demonstrates its ability to modulate transgene expression in a concentration-dependent manner.

Use of human cells: We agree that testing these constructs in human cells is essential for future translational applications and are actively exploring opportunities to evaluate them in patientderived FLS. However, previous studies have shown that Theiler’s virus L* does not inhibit human RNase L (Sorgeloos et al., PLoS Pathog 2013). As a result, it is highly likely that the E3-NSs-L* construct will not function as intended in human systems. Addressing this limitation will be a priority in our future work, where we aim to develop constructs incorporating inhibitors specific to human RNase L to ensure efficacy in human cells.

**Recommendations for the authors:**

**Reviewer #1 (Recommendations for the authors):**
Figure 2c is not indicated.

Thank you for pointing out this error. It has now been corrected in the revised manuscript.

**Reviewer #2 (Recommendations for the authors):**
(1) The Graphical Abstract is a bit confusing; suggest modifying it to represent the study and findings more accurately.

We have revised the graphical abstract to improve clarity and better reflect the study’s design and main findings. Thank you for the suggestion.

(2) The impact of this paper would be greatly improved if these experiments were repeated, at least partially, in human cells. The rationale for mouse cells in vitro is unclear.

The rationale for developing constructs targeting mouse cells is based on our intention to utilize these constructs in mouse models of inflammatory joint pain in future studies.

We recognize that incorporating data from human cells would significantly enhance the translational relevance of our work, and we are actively pursuing collaborations to test these constructs in patient-derived FLS. However, a key component of our saRNA constructs—Theiler’s virus L*—has been shown to inhibit mouse, but not human, RNase L (Sorgeloos et al., PLoS Pathog 2013). Consequently, the E3-NSs-L* polyprotein may not function as intended in human cells. To address this limitation, future work will focus on developing constructs that incorporate inhibitors specific to human RNase L, thereby facilitating more effective translation of our findings to human systems.

(3) The ML336 was only tested at one concentration and works mildly well, but would be more impactful if tested in a dose-response curve.

We have now performed a concentration–response analysis for ML336 (Figure 8a and b), which demonstrates its concentration-dependent effects on transgene expression and saRNA elimination. Thank you for the suggestion.

(4) Overall, there is not a cohesive narrative to the story, instead it comes off as we tried these three different approaches, and they worked in different contexts.

We have revised the graphical abstract, results, and discussion to improve the cohesiveness of the manuscript’s narrative and to better integrate the mechanistic rationale linking the different approaches. We appreciate the feedback.

(5) The title is not supported by the data; the saRNA is still somewhat cytotoxic, immunostimulatory and the antiviral minimally controls transgene expression; suggest making this reflect the data.

We have revised the title to better reflect the scope of the data and the mechanistic focus of the study. The updated title emphasizes the pathways targeted and the outcomes demonstrated, while avoiding overstatement. Thank you for this helpful recommendation.